# Bre1/RNF20 promotes Rad51-mediated strand exchange and antagonizes the Srs2/FBH1 helicases

Guangxue Liu[1,6], Jimin Li[1,6], Boxue He [2], Jiaqi Yan[1], Jingyu Zhao[1], Xuejie Wang[1], Xiaocong Zhao[3], Jingyan Xu[4], Yeyao Wu[5], Simin Zhang[1], Xiaoli Gan[1], Chun Zhou [5], Xiangpan Li[1], Xinghua Zhang [1,3] & Xuefeng Chen [1] ✉

Central to homologous recombination (HR) is the assembly of Rad51 recombinase on single-strand DNA (ssDNA), forming the Rad51-ssDNA filament. How the Rad51 filament is efficiently established and sustained remains partially understood. Here, we find that the yeast ubiquitin ligase Bre1 and its human homolog RNF20, a tumor suppressor, function as recombination mediators, promoting Rad51 filament formation and subsequent reactions via multiple mechanisms independent of their ligase activities. We show that Bre1/RNF20 interacts with Rad51, directs Rad51 to ssDNA, and facilitates Rad51-ssDNA filament assembly and strand exchange in vitro. In parallel, Bre1/RNF20 interacts with the Srs2 or FBH1 helicase to counteract their disrupting effect on the Rad51 filament. We demonstrate that the above functions of Bre1/RNF20 contribute to HR repair in cells in a manner additive to the mediator protein Rad52 in yeast or BRCA2 in human. Thus, Bre1/RNF20 provides an additional layer of mechanism to directly control Rad51 filament dynamics.

Homologous recombination (HR) is an essential mechanism, providing high-fidelity, template-directed repair or tolerance of a plethora of DNA lesions, including single-strand DNA (ssDNA) gaps and double-strand DNA breaks (DSBs), interstrand crosslinks, collapsed replication forks and eroded telomeres[1–6]. As a result, the deregulation of HR can lead to genome instability, cancer, and meiotic defects[2,3,5,7]. In addition, in mice, loss of key HR genes is lethal[7].

HR is initiated with the processing of the 5′-ends of DSBs by nucleases, generating 3′-tailed ssDNA[8]. The 3′-ssDNA is coated by the conserved ssDNA binding protein complex Replication Protein A (RPA). Subsequently, Rad51 recombinase replaces the ssDNA-bound RPA to form the Rad51-ssDNA presynaptic filament, the key intermediate driving HR repair. The Rad51 filament performs homology search and strand invasion of a homologous duplex to form a displacement loop (D-loop)[1–3]. The resolution of the joint recombination molecules accomplishes repair.

During the formation of Rad51-ssDNA filament, excess ssDNA-bound RPA restrains Rad51 nucleation by competing for ssDNA binding sites[9,10]. The mediator protein Rad52 in yeast or BRCA2 in humans plays key roles in facilitating Rad51 nucleation and filament elongation[9–19]. Meanwhile, the Rad51 paralogs Rad55-Rad57 and the Shu complex in yeast and RAD51B-RAD51C in humans also implement accessory roles during this process[7,10,20–25]. However, given the complex nature of various DNA lesions on chromatin, it is unclear whether specific mediator proteins are required to assist the efficient assembly of the Rad51-ssDNA filament for a given type of DNA lesions.

The Rad51 filament is highly vulnerable to the dismantling activities of anti-recombinases. In yeast, the Srs2 helicase disrupts the Rad51-ssDNA filament by triggering Rad51 ATPase activity, leading to

[1]Hubei Key Laboratory of Cell Homeostasis, College of Life Sciences, TaiKang Center for Life and Medical Sciences, Frontier Science Centre of Immunology and Metabolism, Department of Oncology, Renmin Hospital of Wuhan University, Wuhan University, Wuhan, China. [2]Department of Biochemistry and Structural Biology, University of Texas Health Science Center, San Antonio, TX, USA. [3]The Institute for Advanced Studies, Wuhan University, Wuhan, China. [4]Department of Hematology, Nanjing Drum Tower Hospital, the Affiliated Hospital of Nanjing University Medical School, Nanjing, China. [5]School of Public Health, Zhejiang University School of Medicine, Hangzhou, China. [6]These authors contributed equally: Guangxue Liu, Jimin Li. ✉e-mail: xfchen@whu.edu.cn

its disassociation[26,27]. Srs2 prevents excess or untimely recombination events that are toxic to cells[1–3,28,29]. The effect of Srs2 on Rad51 filament can be partially alleviated by the presence of Rad52, Rad55-Rad57, or the Shu complex[10,30–32]. Analogous to yeast Srs2, human BLM, FBH1, PARI, RECQL5, and FIGNL1 can also attenuate HR by disrupting Rad51 filament[33–37]. However, little is known about how their activities are balanced to ensure proper HR in humans.

The yeast ubiquitin ligase Bre1 and its human homolog RNF20, a tumor suppressor, were shown to facilitate DNA replication and repair via promoting histone H2B ubiquitination (H2Bub)[38–47]. In this study, we found that Bre1/RNF20 also functions as a recombination mediator protein in an E3 ligase-independent manner. We show that Bre1/RNF20 physically stimulates Rad51 filament formation and strand exchange and antagonizes the anti-recombinase activities of yeast Srs2 or human FBH1. Notably, in addition to repair the HO endonuclease-induced DSB, the Bre1-mediated mechanism is specifically required for the recombination events induced by bleomycin, a commonly used anticancer drug, and this function cannot be substituted by Rad52. Thus, we revealed a recombination mediator with DNA lesion-specific preference.

## Results

### Bre1 directly interacts with Rad51 in vivo and in vitro

Our recent study showed that RPA recruits the ubiquitin E3 ligase Bre1 to replication forks or DSBs to stimulate local H2Bub, replication, and repair[48]. Interestingly, we noted that disruption of the RPA-Bre1 interaction resulted in increased RPA binding, accompanied by reduced Rad51 loading at DSBs (Supplementary Fig. 1a, b), implying a role of Bre1 in facilitating Rad51 loading on RPA-coated ssDNA. Indeed, we identified Rad51 as a Bre1-interacting partner upon MMS treatment in a mass spectrometry analysis (Fig. 1a). Furthermore, we confirmed their direct interaction by immunoprecipitation or pull-down assay and noted that the interaction could be stimulated by DNA damage (Fig. 1b–d).

Bre1 contains a Rad6-binding domain (RBD, 1–210 amino acids, aa), a RING domain (RD, 615–700 aa), and a linker region harboring coiled coils (211–614 aa) (Fig. 1e). By GST pull-down assay, we found that Bre1 binds Rad51 via the linker region rather than the RBD or RING domain (Fig. 1f). The Bre1 truncation peptides with residues up to 1–496 aa did not bind Rad51, while truncated Bre1 with residues 1–504 aa could bind Rad51 (Fig. 1g, lanes 1–7). Further analysis revealed that the residues near 501, specifically E500 and K502, are important for the interaction (Fig. 1g, lanes 8–14). Simultaneous mutation of the two residues to alanine (*EK2A*) completely abolished the interaction both in vitro and in vivo (Fig. 1g, lanes 13–14; Fig. 1h). Thus, we identified E500 and K502 as the key Bre1 residues for mediating the Bre1-Rad51 interaction.

### Bre1 interacts with Rad51 to promote DNA damage response, Rad51 loading, and HR repair

We first tested the role of the Bre1-Rad51 interaction in the DNA damage response. We observed that disruption of the interaction by the *EK2A* mutation caused hypersensitivity to the bleomycin-family DNA-damaging agents that can induce complex DNA lesions, including ssDNA gaps and DSBs (Fig. 2a)[49–51]. Upon short exposure to phleomycin, the *EK2A* mutant recovered slower than the wild-type (WT) cells, as reflected by the formation and disappearance of Rad52-GFP foci, an indicator of DNA lesion (Fig. 2b, c). Consistently, the mutant cells had a reduced survival rate following short phleomycin treatment (Fig. 2d). These results suggest that the mutant has an impaired ability to repair DNA damage. Next, we tested the efficiency of DSB repair by ectopic recombination using a system where a single HO cut is generated on chromosome V that can be repaired by donor sequence from chromosome III (Fig. 2e)[52]. About 85% of WT cells completed the repair and survived, while only ~60% of *EK2A* cells and 47% of *bre1Δ* survived

(Fig. 2f). Consistently, Rad51 loading at DSB ends was impaired in *EK2A* cells (Fig. 2g). However, this is not due to any changes in Rad51 or Bre1 protein levels (Supplementary Fig. 2a, b).

Cells lacking Srs2 are susceptible to DNA-damaging agents, probably due to the accumulation of unresolved recombination intermediates[26,27]. Interestingly, the *EK2A* mutation markedly suppressed the DNA damage sensitivity of *srs2Δ* cells and increased the survival rate of *srs2Δ* cells from DSB repair by ectopic recombination fivefold (Fig. 2a, h). Thus, the Bre1-Rad51 interaction is required for proper DNA damage response, Rad51 loading, and HR repair and exhibits a strong genetic interaction with Srs2.

### The Bre1-Rad51 interaction facilitates HR repair in a manner independent of the E3 ligase activity of Bre1

It has been shown that Bre1-mediated H2Bub promotes DSB repair by HR and that both RD and RBD domains of Bre1 are important for H2Bub[47,53]. Notably, the Bre1 truncation mutant lacking the RBD (*bre1-RBDΔ*) or RING domain (*bre1-RDΔ*) had a higher level of survival from DSB repair by HR (~60%) than the *bre1Δ* null mutant (47%) (Fig. 2f), suggesting that Bre1 has an H2Bub-independent role in HR. On the other hand, although the *EK2A* mutant exhibited a defect in HR, it had normal levels of H2Bub in unperturbed or hydroxyurea-stressed conditions. Moreover, the enrichment of Bre1 and H2Bub at DSBs was proficient in *EK2A* cells (Supplementary Fig. 2c–e). Finally, we noted that the EK2A mutant protein exhibited normal E3 ligase activity toward H2B in vitro (Supplementary Fig. 2f, g). Together, these results indicate that the HR defect in *EK2A* cells was independent of the ubiquitin ligase activity of Bre1.

### Bre1 stimulates Rad51 assembly on ssDNA

Next, we examined whether Bre1 can directly regulate Rad51-ssDNA filament formation. We observed that the presence of Bre1 enhanced the resistance of the Rad51-ssDNA complex to high salt (250–300 mM NaCl) (Supplementary Fig. 3a, b). Notably, the addition of Bre1 markedly stimulated Rad51 binding to ssDNA coupled to streptavidin-coated magnetic beads (Fig. 3a–c, lanes 4–6). Although Bre1 alone did not exhibit detectable ssDNA binding ability (lane 3), it was captured in the complex by ssDNA pull down when Rad51 was present (lanes 5–6), implying that Bre1 can bind the Rad51-ssDNA filament. In support of the above results, incubation of Rad51-ssDNA with wild-type Bre1 but not the EK2A mutant protein led to the formation of shifted protein-ssDNA complexes stacked in gel wells, as measured by an electrophoretic mobility shift assay (EMSA) (Fig. 3d, lanes 1–8).

We further employed single-molecule magnetic tweezers (MT) to monitor the effect of Bre1 on Rad51-ssDNA assembly. We labeled 12.5 k-nt ssDNA molecules with digoxigenin and biotin at the 5′- and 3′-ends, respectively, and stretched ssDNA molecules using MT at 8 pN, 20 °C, and pH 7.5 (Fig. 3e). The formation of Rad51-ssDNA filaments results in an extension in ssDNA length that can be monitored using single-molecule MT[54,55]. For each condition, the average values from multiple ssDNA molecules were calculated and plotted as a time-extension curve. The addition of Bre1 alone did not change ssDNA length (Supplementary Fig. 4a). In contrast, the addition of Rad51 to ssDNA led to a striking extension of ssDNA, and it took over 30 min to saturate the ssDNA (Fig. 3f). Notably, the inclusion of Bre1 in the reaction drastically accelerated the assembling process, shortening the time to ~5 min (Fig. 3f). Thus, Bre1 plays a critical role in stimulating the dynamic assembly of Rad51 with ssDNA. Interestingly, a Bre1 peptide (400–504 aa, pep104) containing the Rad51-interacting motif was sufficient to stimulate the reaction (Fig. 3f and Supplementary Fig. 4b). Hydrolysis of ATP by Rad51 ATPase leads to the disassociation of Rad51 from ssDNA[26,27]. We found that Bre1 did not inhibit Rad51 ATPase activity (Supplementary Fig. 5a). Thus, we conclude that Bre1 can directly bind Rad51-ssDNA filament and stimulate the nucleation or elongation of the filament in an E3 ligase-independent manner.

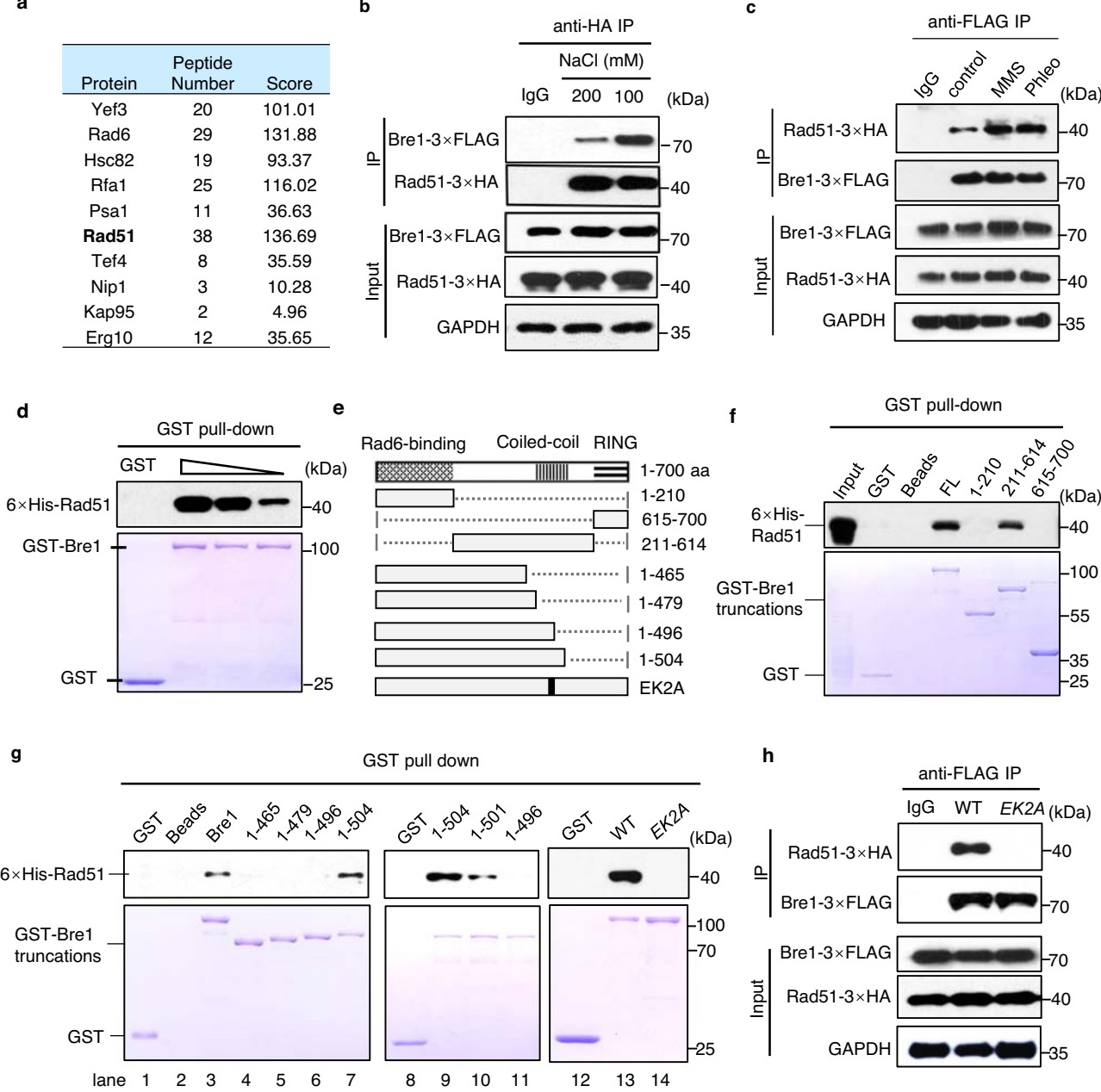

**Fig. 1 | Bre1 interacts with Rad51 in vivo and in vitro. a** Table showing the Bre1-interacting proteins identified by mass spectrometry analysis. The unique peptide number for each protein is indicated. The score is an indicator of reliability. **b**, **c**, **h** Immunoprecipitation and Western blot analysis of the interaction between Rad51-3xHA and Bre1-3xFLAG or bre1-EK2A-3xFLAG at indicated conditions. GAPDH was used as a loading control. Yeast cells were treated with phleomycin (20 μg/ml) for 2 h or MMS (0.1%) for 1 h before collection. Asynchronized cells were used as control. **d**, **f**, **g** GST pull-down assays showing the interaction between the GST-tagged WT or mutant Bre1 and 6xHis-Rad51. GST was used as a negative control. The bottom panel shows Coomassie blue staining of GST-tagged proteins used for the experiment. The upper panel is the Western blot showing the levels of 6xHis-Rad51. The sizes of proteins are indicated by the molecular-weight markers. **e** Scheme showing the full-length or truncated Bre1 proteins used for pull-down assays. Dot lines represent deleted regions, while the gray bars represent the truncated Bre1 proteins. The positions for these truncations are indicated. The Rad51-interacting motif of Bre1 is marked by the black bar. Source data are provided as a Source Data file.

## Bre1 promotes the replacement of ssDNA-bound RPA by Rad51

We then tested whether Bre1 helps overcome the inhibitory effect of RPA on Rad51 nucleation using an ssDNA affinity pull-down assay (Fig. 3g). In the absence of RPA, WT Bre1 but not the EK2A mutant protein stimulated Rad51 binding on ssDNA (Fig. 3h, i, lanes 4–6). The addition of RPA markedly inhibited Rad51 binding to ssDNA, and this inhibition was largely alleviated by the inclusion of Bre1 (lanes 7–8). To confirm this result, we monitored the replacement of ssDNA-bound RPA by Rad51 using single-molecule MT (Fig. 3j, k)[54,55]. First, a sufficient

amount of RPA (300 nM) was added to saturate ssDNA (Fig. 3k and Supplementary Fig. 4c), and then the free RPA in the flow cell was rinsed with the assembling buffer. Subsequently, Rad51(750 nM) was added with or without Bre1 to the reaction. Rad51 alone could fully replace RPA in ~75 min, while the inclusion of Bre1 or the peptide pep104 in the reaction markedly accelerated the replacement to ~30 min (Fig. 3k). These results demonstrate that Bre1 can act as a recombination mediator, facilitating Rad51 nucleation and replacement of the ssDNA-bound RPA.

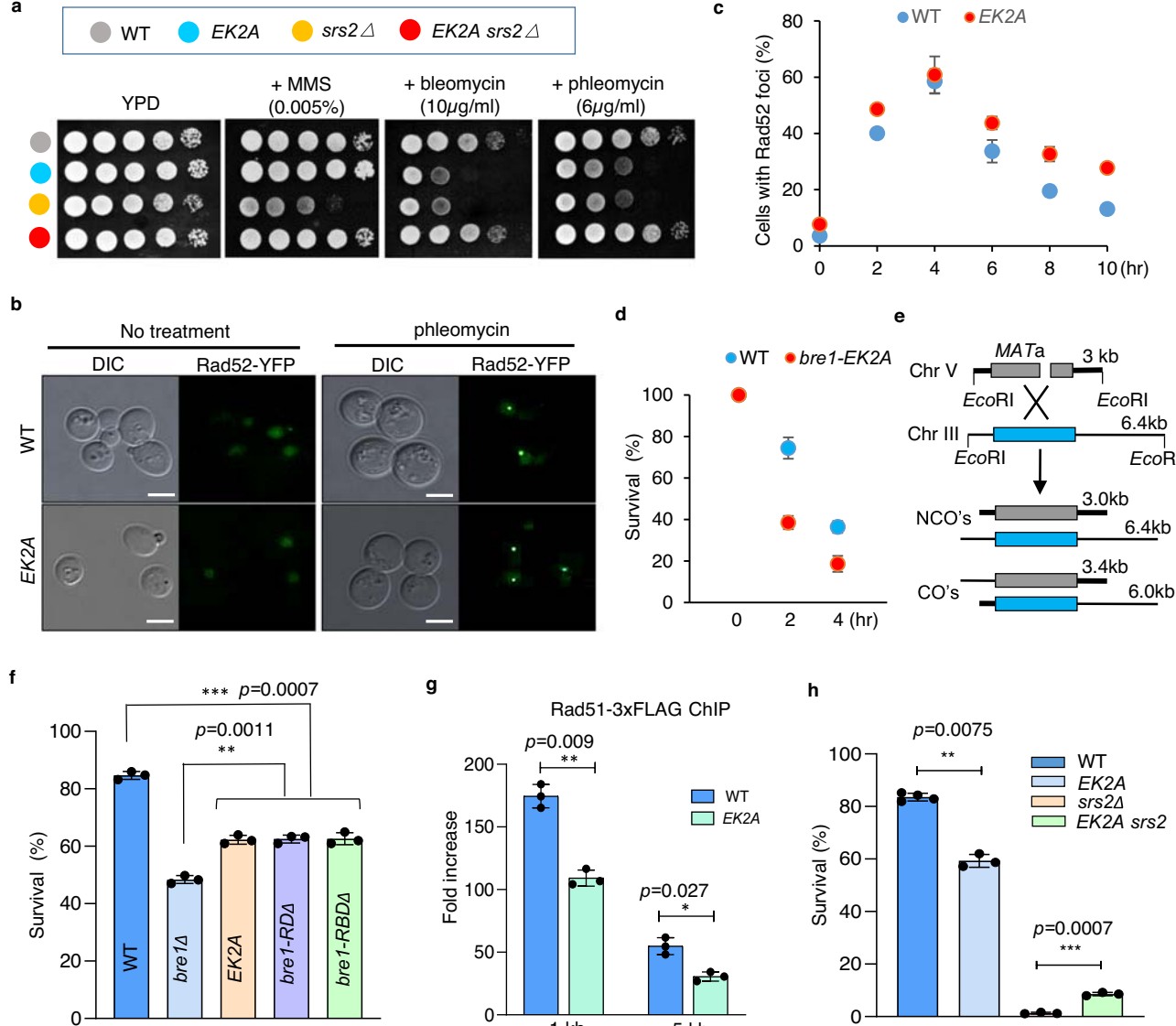

**Fig. 2 | The interaction between Bre1 and Rad51 promotes Rad51 loading, HR repair, and DNA damage response. a** Spotting assays showing the DNA damage sensitivity for indicated cells at indicated drug concentrations. The colorful dots represent different strains. **b**, **c** Microscopy analysis and quantification of cells with Rad52-YFP foci upon or after phleomycin treatment. Cells were treated with 20 μg/ml phleomycin for 2 h and then were released into media without the drug. Rad52-YFP foci were monitored at indicated time points. A representative image at 8 h is presented. Scale bar, 3 μm. Values represent the mean ± SD of three independent experiments (n = 3). **d** The survival rate for the WT or *EK2A* cells after short phleomycin treatment (20 μg/ml). Values represent the mean ± SD of three independent experiments (n = 3). **e** Scheme showing an ectopic recombination system. CO crossover, NCO non-crossover. **f**, **h** The survival rate of DSB repair by ectopic recombination for indicated cells. Values represent the mean ± SEM of three independent experiments (n = 3). **g** ChIP-qPCR analysis of Rad51-3xFLAG loading at 1 or 5 kb away from the HO cut site. Values represent the mean ± SEM of three independent experiments (n = 3). Samples were collected 4 h post galactose induction. Statistical analysis was calculated with the Student *t*-test (two-tailed). *p < 0.05, **p < 0.01, ***p < 0.001. Source data are provided as a Source Data file.

## Bre1 stimulates Rad51-mediated strand exchange

Next, we examined whether Bre1 affects Rad51-mediated strand exchange reaction in vitro using a previously described method[56]. We preincubated RPA with ssDNA (80 nt) before adding Rad51 and then initiated the reaction by adding a Cy3-labeled homologous duplex DNA (40 bps). The Rad51-mediated homologous pairing and strand exchange reactions generate a Cy3-labeled hybrid duplex DNA (Fig. 3l, m, lanes 3–4). We noted that the inclusion of Bre1 significantly increased the exchange products (Fig. 3m, n, lanes 5–6). Thus, Bre1 could directly stimulate the Rad51-mediated strand exchange reaction.

## Bre1 restrains Rad51 binding to dsDNA

Rad51 binds both ssDNA and dsDNA, and its binding to duplex DNA strongly inhibits DNA strand exchange[18,57]. We found that incubation of

Rad51 with linear dsDNA resulted in the formation of the super-shifted Rad51-dsDNA complexes stacked in gel wells (Supplementary Fig. 6a, lanes 1–3). Interestingly, the addition of Bre1 together with Rad51 to the reaction reduced the complex formation (lanes 4–5). However, when Rad51 was preincubated with dsDNA to form the Rad51-dsDNA complex, the addition of Bre1 could not displace Rad51 from dsDNA anymore (lanes 6–7). These results suggest that Bre1 sequesters Rad51 to prevent its binding to dsDNA. As a control, the addition of Rad52 together with Rad51 to the reaction did not impair the complex formation (Supplementary Fig. 6b, lanes 6–8). To confirm this result, we performed a competitive DNA binding assay. The presence of dsDNA impaired Rad51 binding to the ssDNA coupled to streptavidin-coated magnetic beads (Supplementary Fig. 6c–e, lanes 2–3). The inclusion of Bre1 repressed the competitive effect of dsDNA (lanes 4–5). These results indicate that Bre1

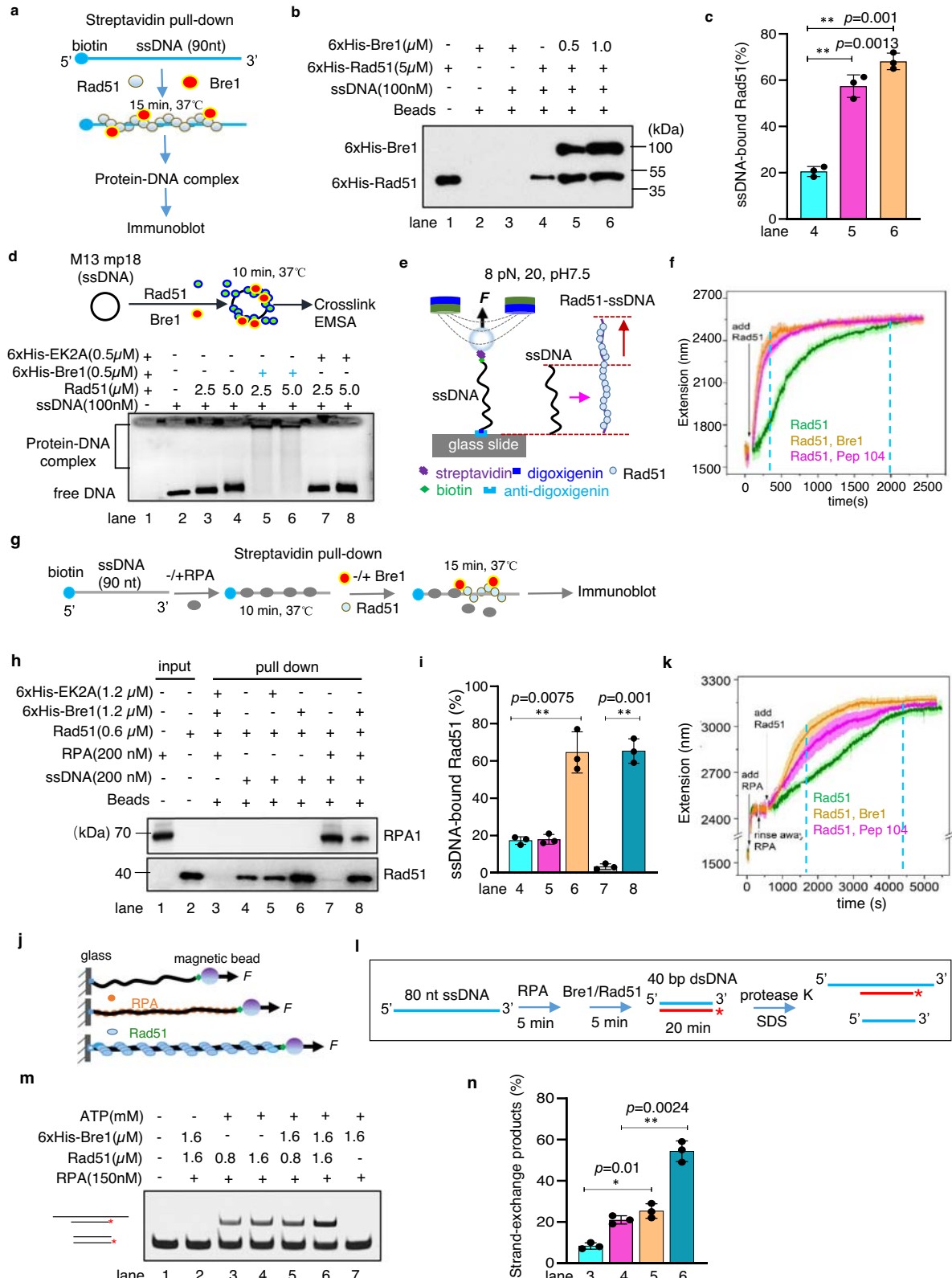

promotes Rad51 loading to ssDNA while sequestering Rad51 to restrain its binding to dsDNA, as reported for human BRCA2[14].

## Bre1 protects Rad51-ssDNA filaments against Srs2 dismantling activity

Srs2 prevents unscheduled or excess recombination by disrupting the Rad51-ssDNA presynaptic filament[26,27]. Indeed, we observed that Rad51 binding to biotin-labeled ssDNA (90 nt) was significantly impaired upon the addition of Srs2 (1.5 µM) (Fig. 4a–c, lanes 4–5). Notably, the disrupting effect of Srs2 on Rad51-ssDNA was fully repressed by the inclusion of Bre1 (Fig. 4b, c, lanes 6–8). Consistently, the presence of Srs2 attenuated Rad51-mediated strand exchange products in a dose-dependent manner (Fig. 4d, e, lanes 2–4 and Supplementary Fig. 7a, b, lanes 3–5), while the inclusion of Bre1 or the peptide pep104 in the

**Fig. 3 | Bre1 promotes Rad51-ssDNA filament assembly and strand exchange in vitro. a, g** Scheme showing the ssDNA pull-down assays. **b,** An ssDNA pull-down assay followed by Western blot analysis showing the effect of Bre1 on Rad51 binding to ssDNA. **c** Quantification of the relative amount of ssDNA-bound Rad51 presented in (**b**). Values represent the mean ± SD of three independent experiments ($n = 3$). **d** An electrophoretic mobility shift assay showing the effect of Bre1 or EK2A protein on Rad51 binding to M13 mp18 circular ssDNA. **e** A single-molecule magnetic twister system for monitoring the kinetics of Rad51 assembling on ssDNA. The ssDNA was stretched using MT at 8 pN, 20 °C, and pH 7.5. **f** Single-molecule MT analysis showing the effect of Bre1 or the peptide pep104 on Rad51 assembling with ssDNA. 150 nM of Rad51 and 150 nM of Bre1 or Pep104 were used for experiments. **h, i** An ssDNA pull-down assay and quantification showing the effect of Bre1 or EK2A on the replacement of RPA by Rad51. Values represent the mean ± SD of three independent

experiments ($n = 3$). **j, k** Single-molecule MT analysis of the effect of Bre1 or Pep104 on the replacement of ssDNA-bound RPA by Rad51. ssDNA was initially saturated with 300 nM of RPA. Subsequently, 750 nM of Rad51 with or without Bre1(750 nM) or Pep104(750 nM) was used for the reaction. The average (dark-colored) and SD (lighter-colored) of the time courses at each condition in (**f**) and (**k**) were obtained from three independent molecules ($n = 3$). **l** Scheme showing the assay for measuring the strand exchange reaction. **m** An EMSA assay showing the effect of Bre1 on Rad51-mediated strand exchange. **n** Quantification of the relative amount of the strand exchange products showed in (**m**). Values represent the mean ± SD of three independent experiments ($n = 3$). Statistical analysis was calculated with the Student *t*-test (two-tailed). *$p < 0.05$, **$p < 0.01$. Source data for panels are provided as a Source Data file.

reaction completely suppressed the disrupting effect of Srs2 on the formation of these products (Fig. 4d, e, lanes 5–6 and Supplementary Fig. 7a, b, lanes 6–7). Thus, Bre1 directly antagonizes the dismantling activity of Srs2 on Rad51 filament formation and strand exchange.

## Bre1 dismantles Srs2 from ssDNA in vitro

We then tested how Bre1 may counteract Srs2. The deletion or mutation of *BRE1* did not change the Srs2 protein level or the interaction between Srs2 and Rad51 (Supplementary Fig. 8a, b, lanes 7, 10, and 11). In addition, the presence of Bre1 did not alter the ATPase activity of Srs2 (Supplementary Fig. 8c). Next, we tested whether Bre1 impacts Srs2 binding on ssDNA (Fig. 4f). The binding of Srs2 to M13 mp18 ssDNA resulted in the formation of high molecular-weight complexes that displayed retarded mobility on an electrophoresis gel (Fig. 4g, lanes 2–4). Surprisingly, the addition of Bre1 to the pre-formed Srs2-ssDNA complexes led to the disassociation of Srs2 from ssDNA (Fig. 4g, lanes 5–7). To confirm this result, we performed an ssDNA pull-down assay and observed that Bre1 restrained Srs2 binding to ssDNA irrespective of the timing of Bre1 addition, no matter before (Fig. 4h, i, lanes 4–6) or after the formation of the Srs2-ssDNA complex (lanes 7–9). Notably, Bre1 can also efficiently dismantle Srs2 from Y-shape or 3'-flap DNA molecules but not from dsDNA (Supplementary Fig. 9a, b). Interestingly, the EK2A or the RING-deleted Bre1 (RINGΔ) mutant protein can also unload Srs2 from ssDNA (Supplementary Fig. 9c, d, lanes 3–6). Together, these results indicate that Bre1 is capable of displacing Srs2 from ssDNA independently of its ligase activity.

## Bre1 interacts with Srs2 to regulate Rad51 and Srs2 loading and HR repair

We observed that Bre1 interacted with Srs2 in vivo in a manner dependent on DNA since benzonase treatment attenuated the interaction (Fig. 5a). Consistently, purified GST-Srs2 binds 6xHis-Bre1 only when ssDNA was included (Fig. 5b, lanes 6, 7), suggesting that the interaction occurs on chromatin. However, they did not interact when dsDNA was included (Fig. 5b, lanes 8, 9). Importantly, deletion of *BRE1* or disruption of the Bre1-Rad51 interaction by the EK2A mutation led to increased Srs2 loading at DSBs (Fig. 5c). Thus, the direct Bre1-Rad51 interaction is required for Bre1 to displace Srs2 from ssDNA in the chromatin context.

Next, we determined the Bre1 domain required for the interaction with Srs2. We noted that the Bre1 truncation with residues 1–501 aa, but not the one with residues 1–485 aa, retained the ability to interact with Srs2 (Fig. 5d, e, lanes 1–3). Notably, the simultaneous mutation of D484 and K486 to alanine (*DK2A*) completely abolished the interaction (Fig. 5e, lanes 6–7). In line with the role of Bre1 in antagonizing Srs2, disruption of the Bre1-Srs2 interaction resulted in increased Srs2 loading at DSBs, accompanied by reduced Rad51 binding (Fig. 5c, f). Consequently, both *DK2A* and *EK2A* mutants displayed reduced HR survival, repair kinetics, crossover level, and resistance to phleomycin or zeocin (Fig. 5g, h and Supplementary Fig. 10a–c). Therefore, the

Bre1-Srs2 interaction is required to balance the loading of Rad51 and Srs2 and promote proper HR repair.

## Bre1 acts additively to Rad52 and Rad55-Rad57

Rad52 is the key mediator facilitating Rad51 nucleation on RPA-coated ssDNA, while the Rad55-Rad57 complex plays an auxiliary role epistatic to Rad52[16–18,20,21]. The mediator function of Rad52 relies on its interaction with Rad51 and RPA[16,17]. The Rad52-Y376A mutation abolishes the Rad52-Rad51 interaction and therefore disrupts its mediator function[32]. Both *rad52-Y376A* and *rad57Δ* single mutants exhibited decreased ectopic recombination repair of the HO-induced DSBs, while additional mutation of *EK2A* in either mutant further reduced the survival rate (Supplementary Fig. 11a). Thus, Bre1 acts additively to Rad52 or Rad55-Rad57 in promoting HR repair. Interestingly, an excess of Rad52 but not the vector itself can fully rescue the HR defect in *EK2A* mutant cells, while an excess of Bre1 can also partially rescue the HR defect of *rad52-Y376A* cells (Supplementary Fig. 11b). Thus, Rad52 and Bre1 can at least partially compensate for the loss of the other in repairing the HO-induced DSBs. However, overexpression of Rad52 cannot rescue the sensitivity of *EK2A* cells to phleomycin at all (Supplementary Fig. 11c), implying that Bre1 is indispensable to mediate the recombination repair of the complex DNA lesions induced by phleomycin that cannot be accomplished by Rad52 alone.

## Human RNF20 promotes hRad51-ssDNA assembly and strand exchange and counteracts the FBH1 helicase

To examine whether the Bre1-mediated mechanism is conserved in humans, we tested the interaction between hRad51 and RNF20, the human homolog of Bre1. Indeed, we found that RNF20 co-immunoprecipitated with endogenous hRad51 in HEK293T cells and interacted with recombinant hRad51 in vitro (Fig. 6a, b). Moreover, we found that RNF20 interacted with hRad51 via the N-terminal fragment RNF20N (1-300 aa) that does not contain the RING domain (Supplementary Fig. 12a). Since the full-length recombinant RNF20 is unstable, we used the RNF20N fragment for the subsequent studies. Importantly, using an ssDNA pull-down assay, we found that the addition of RNF20N together with hRad51 stimulated hRad51 binding to ssDNA in a dose-dependent manner (Fig. 6c, d, lanes 4–6). As a control, RNF20N itself did not bind ssDNA (lane 3). Indeed, the addition of RNF20N accelerated the dynamic assembly of hRad51 with ssDNA, as monitored by single-molecule MT (Fig. 6e and Supplementary Fig. 4d, e). The stimulation was not through inhibiting hRad51 ATPase activity since the addition of RNF20N did not affect hRad51 ATPase activity (Supplementary Fig. 5b). Like Bre1, RNF20N was also captured by ssDNA pull down when hRad51 was present (Fig. 6c, d, lanes 4–6), suggesting that RNF20 associates with the hRad51-ssDNA filament. Consistently, the addition of RNF20N to hRad51 and ssDNA led to retarded mobility of the Rad51-ssDNA complexes on gels (Supplementary Fig. 12b).

Next, we explored the role of RNF20 in promoting hRad51 nucleation on hRPA-coated ssDNA. We found that hRad51 itself was inefficient in replacing the hRPA bound on ssDNA, as revealed by the

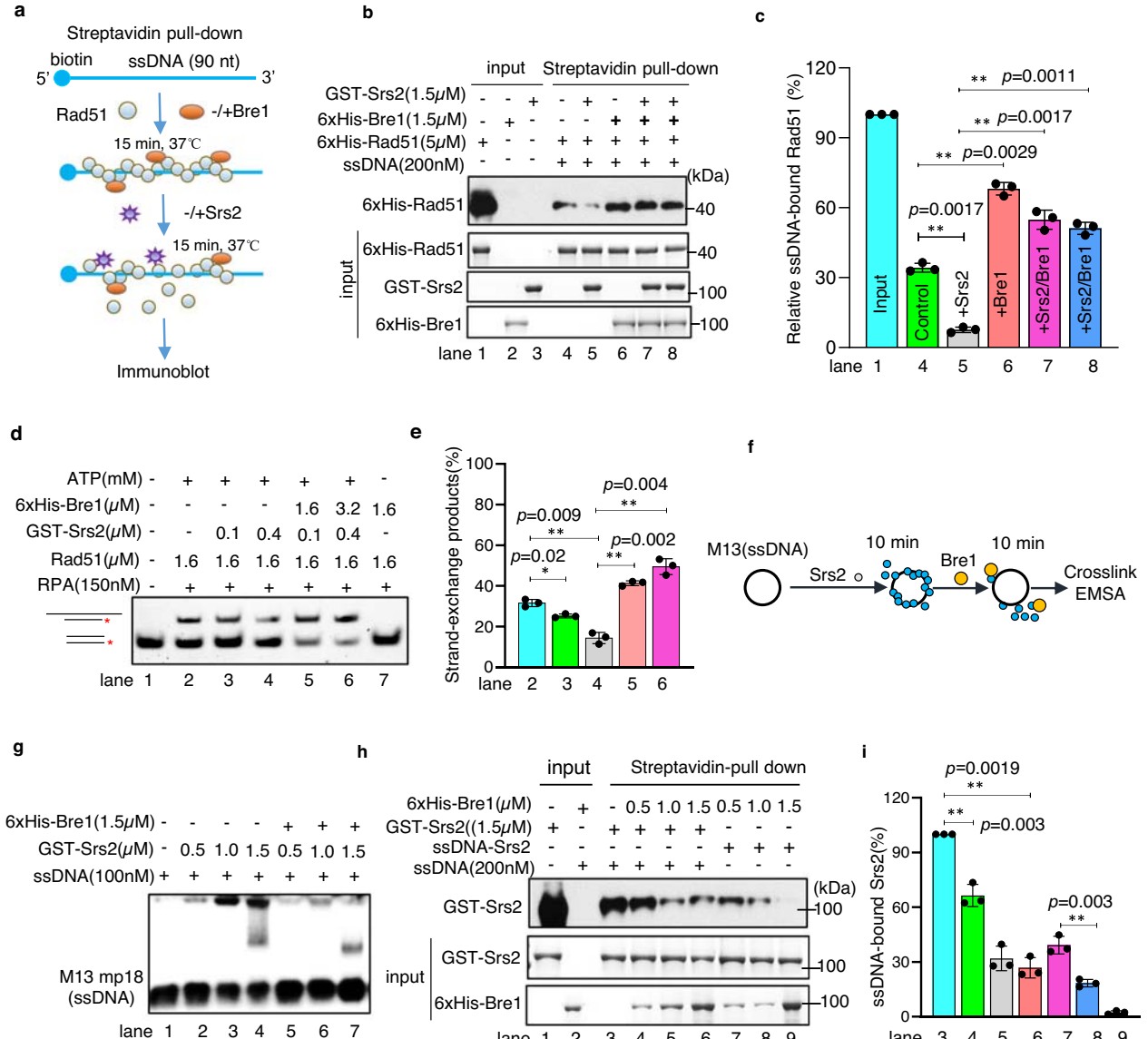

**Fig. 4 | Bre1 protects Rad51 filament from Srs2 dismantling activities via displacing Srs2 from ssDNA. a** Scheme showing an ssDNA pull-down assay. 200 nM of biotin-labeled ssDNA(90 nt) was immobilized to streptavidin-coated beads by incubating at room temperate for 30 min. Proteins were added in the indicated order. **b** An ssDNA pull-down assay followed by Western blot analysis indicating the role of Bre1 in antagonizing the effect of Srs2 on Rad51 filament. Protein and ssDNA concentrations are indicated. **c** Quantification of the relative amount of ssDNA-bound Rad51 showed in (**b**). Values are the mean ± SD of three independent experiments (*n* = 3). **d** An EMSA assay showing the role of Bre1 or Srs2 in Rad51-mediated strand exchange reaction. Protein concentrations are indicated.

**e** Quantification of the relative amount of strand exchange products presented in (**d**). Values are the mean ± SD of three independent experiments (*n* = 3). **f, g** Scheme and EMSA showing the effect of Bre1 on the binding of Srs2 on ssDNA (M13 mp18, 100 nM). Proteins at indicated concentrations were added following the order indicated in (**f**). **h** ssDNA pull-down followed by Western blot analysis showing the effect of Bre1 on the binding of Srs2 to ssDNA (90 nt, 200 nM). **i** Quantification of the relative amount of ssDNA-bound Srs2 showed in (**h**). Values are the mean ± SD of three independent experiments (*n* = 3). Statistical analysis was calculated with the Student *t*-test (two-tailed). *$p < 0.05$, **$p < 0.01$. Source data are provided as a Source Data file.

ssDNA pull-down assay (Fig. 6f, g, lane 5). Notably, the addition of RNF20N could efficiently stimulate hRad51 assembly on hRPA-coated ssDNA in a dose-dependent manner (Fig. 6f, g, lanes 5–8). This result was confirmed by the single-molecule MT studies (Fig. 6h and Supplementary Fig. 4f). These results indicate that RNF20 possesses recombination mediator activity. Like Bre1, RNF20N also suppressed hRad51 binding to dsDNA when it was added together with hRad51 (Supplementary Fig. 12c, lanes 2–4), yet it could not dismantle hRad51 from the pre-formed hRad51-dsDNA complexes (lanes 5–6). Thus, like Bre1, RNF20 also plays a role in sequestrating hRad51 to limit its binding to dsDNA. Notably, we observed that the inclusion of RNF20N stimulated the hRad51-mediated strand

exchange reaction (Fig. 6i, j, lanes 2–4). Together, these results indicate that RNF20 directly promotes hRad51 nucleation and replacement of hRPA and stimulates Rad51-mediated strand exchange via interacting with hRad51.

Analogous to yeast Srs2, the human UvrD family helicase FBH1 can also disrupt the hRad51-ssDNA filament that involves its ssDNA translocation activity[37]. Indeed, the addition of human FBH1 impaired the formation of strand exchange products (Fig. 6k, l, lanes 3–5). However, the inclusion of RNF20N repressed the effect of FBH1, suggesting that RNF20 could counteract the dismantling activity of FBH1 on the hRad51 filament (Fig. 6k, l, lanes 5–7). In support of this notion, the ectopically expressed HA-FBH1

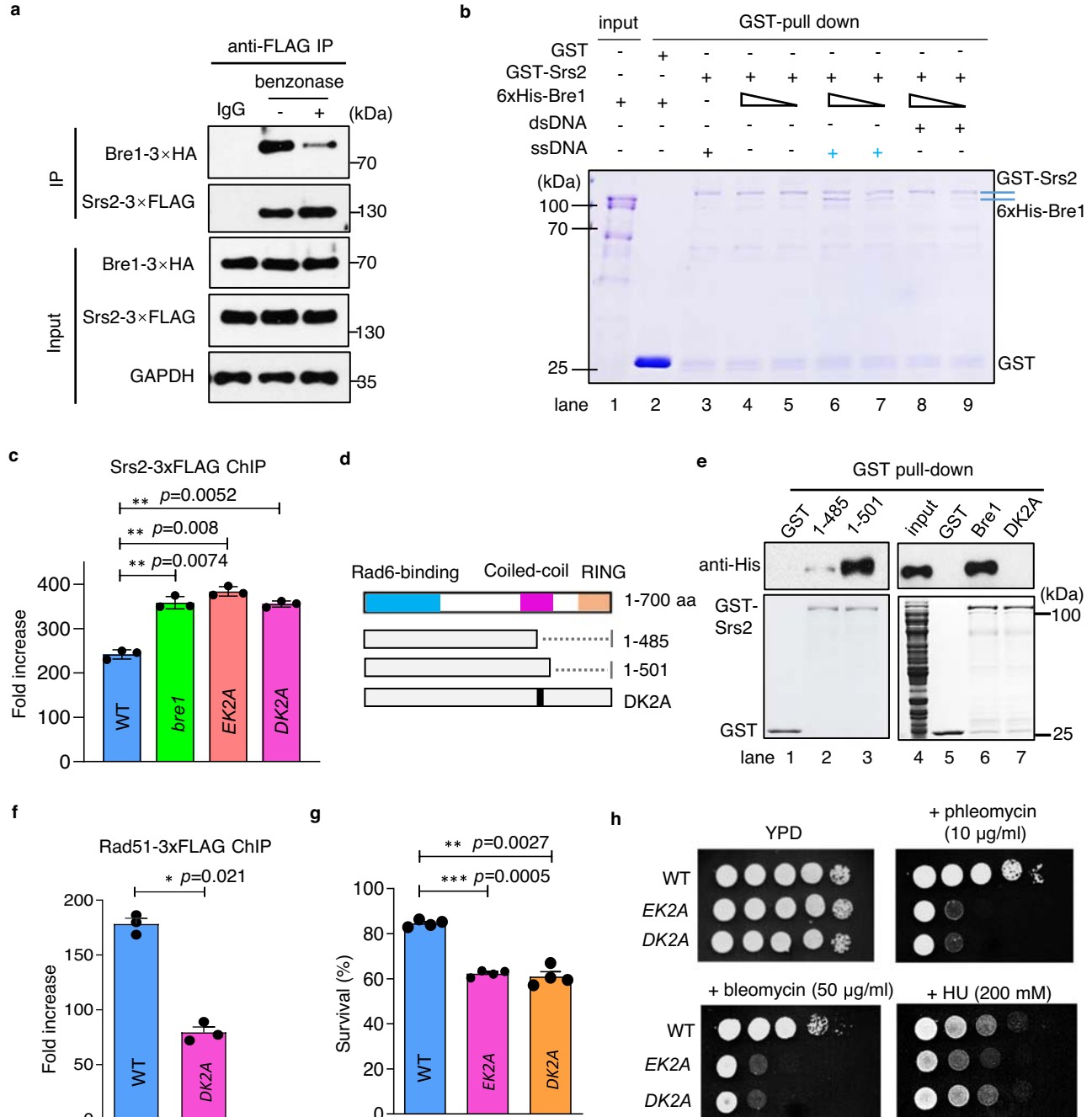

**Fig. 5 | Bre1 physically counteracts Srs2 to facilitate Rad51 loading and HR repair. a** Immunoprecipitation(IP) followed by Western blot analysis showing the interaction between Srs2-3xFLAG and Bre1-3xHA. IgG was used as a mock IP. GAPDH serves as a loading control. **b** A GST pull-down assay indicating the interaction between 6xHis-Bre1 and GST-Srs2. GST was used as a control. The products were resolved on SDS-PAGE followed by Coomassie blue staining. **c**, **f** ChIP analysis of Srs2-3xFLAG or Rad51-3xFLAG loading at 1 kb away from the DSB 4 h after break induction. Values in (**c**) and (**f**) are the mean ± SEM of three independent experiments (*n* = 3). **d** Scheme showing the WT, truncated or mutated Bre1 protein used for GST pull-down assay. The Bre1 motifs are indicated by the colorful bars. **e** GST pull-down assay and Western blot analysis of the interaction between GST-Srs2 and 6xHis-tagged WT, truncated or mutated Bre1 protein. The amount of GST-Srs2 used for experiments is indicated by Coomassie blue staining (lower panel). The product was detected by Western blot with an anti-His antibody (upper panel). **g** The survival rate of DSB repair by ectopic recombination in indicated strains. Values are the mean ± SEM of three independent experiments (*n* = 3). **h** DNA damage sensitivity test for indicated strains. Drug concentrations are indicated. Statistical analysis was calculated with the Student *t*-test (two-tailed). *$p < 0.05$, **$p < 0.01$, ***$p < 0.001$. Source data are provided as a Source Data file.

interacted with FLAG-RNF20, and so did the endogenous FBH1 and RNF20, as revealed by immunoprecipitation (Fig. 6m). The GST pull-down assay confirmed their direct interaction in vitro (Fig. 6n). Thus, Bre1/RNF20 acts as a direct HR regulator promoting hRad51 filament formation and strand exchange while counteracting the anti-recombinase activities of Srs2 or FBH1.

## The ligase-independent roles of RNF20 in promoting hRad51 loading and HR repair

It has been reported that RNF20 facilitates HR repair via promoting histone H2B ubiquitination that requires its RING domain[43,44,58–60]. We attempted to examine whether the above described E3 ligase-independent roles of RNF20 contribute to HR repair. Since the

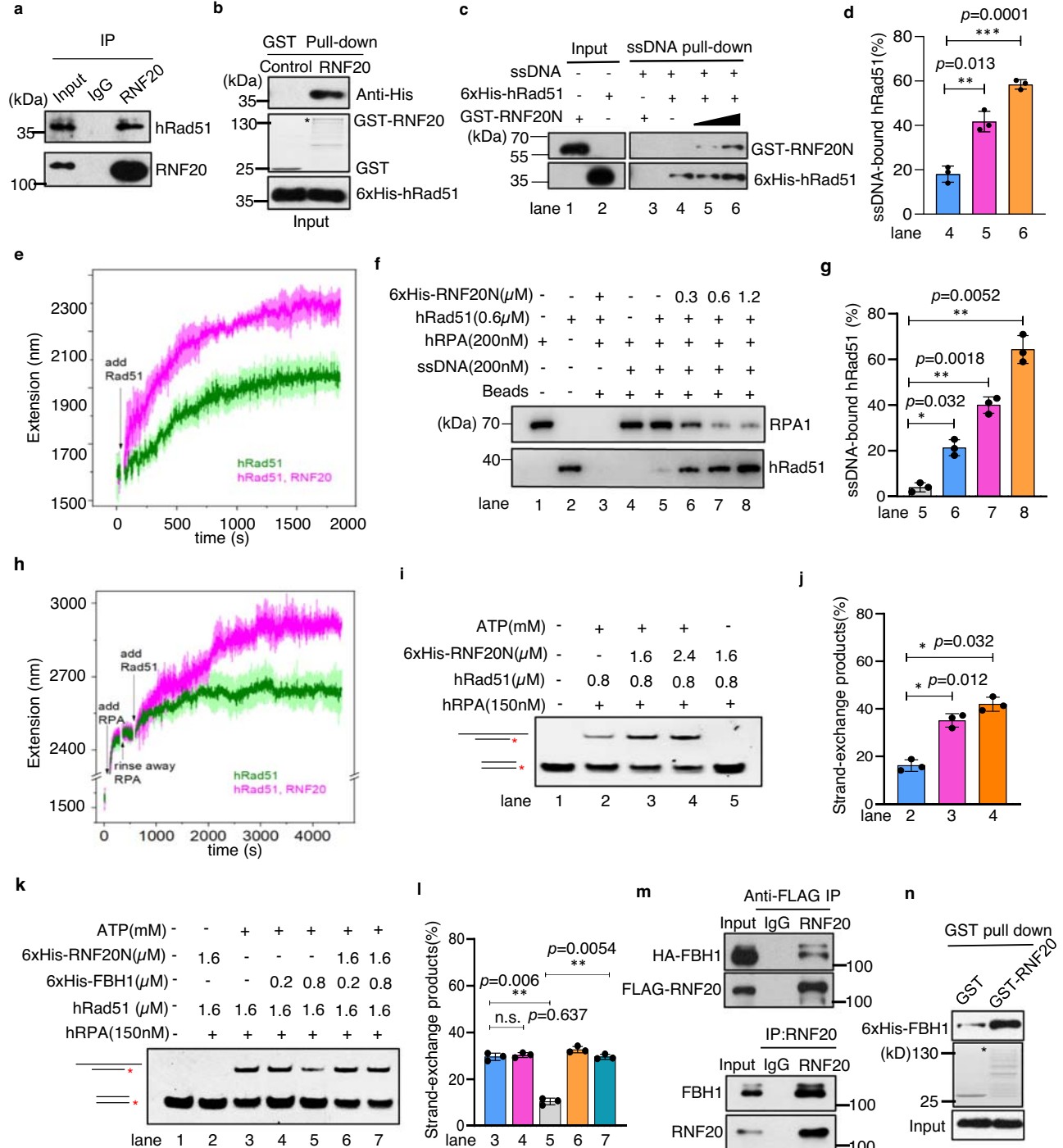

**Fig. 6 | Human RNF20 promotes hRad51-mediated strand exchange and counteracts the disrupting activities of FBH1.** a Immunoprecipitation showing the interaction between endogenous RNF20 and hRad51 in Hela cells. b GST pull-down showing the interaction between GST-RNF20 and 6xHis-hRad51. GST or GST-RNF20 was assessed by Coomassie blue staining. 6xHis-hRad51 was detected by Western blot. The asterisk denotes the full-length GST-RNF20. c, d An ssDNA pull-down assay and quantification showing the effect of RNF20N on hRad51 binding to ssDNA. ssDNA-bound proteins were detected by Western blot with the anti-GST or anti-His antibody. e, h Single-molecule MT monitoring the role of RNF20N on hRad51-ssDNA assembly (e) or on hRad51 replacement of ssDNA-bound hRPA (h). The average (dark-colored) and SD (lighter-colored) of the time courses at each condition were obtained from three independent molecules (*n* = 3). 150 nM of hRad51 and RNF20N were used for the experiment in (e). For testing hRPA replacement by hRad51, ssDNA was initially saturated with 400 nM of hRPA.

Subsequently, 1 μM of hRad51 with or without RNF20N(400 nM) was used for the reaction in (h). f, g An ssDNA pull-down assay and quantification showing the effect of RNF20 on the replacement of ssDNA-bound hRPA by hRad51. i, j Analysis of the effect of RNF20 on the strand exchange reaction by EMSA. Quantification of the relative amount of strand exchange products is presented in (j). k, l EMSA and quantification showing the role of RNF20 in counteracting the effect of FBH1 on hRad51-mediated strand exchange. m Immunoprecipitation indicating the interaction between the ectopically expressed FLAG-RNF20 and HA-FBH1 (upper panel) or between the endogenous RNF20 and FBH1 (lower panel). n GST pull-down assay showing the interaction between GST-RNF20 and 6xHis-FBH1 in vitro. The asterisk indicates the full-length GST-RNF20 protein. Values in **d**, **g**, **j**, **l** are the mean ± SD of three independent experiments (*n* = 3). Statistical analysis was calculated with the Student *t*-test (two-tailed). n.s. no significance; *$p < 0.05$, **$p < 0.01$, ***$p < 0.001$. Source data are provided as a Source Data file.

N-terminal segment of RNF20 can interact with hRad51, RAD6, and RNF40, so we could not specifically discern the role of the RNF20-hRad51 interaction in HR repair by simply deleting or overexpressing the N-terminal segment of RNF20[61]. Therefore, we chose to assess its ligase-independent roles using the *RNF20 RING*-deleted (*RD∆*) allele. Compared to the control, cells stably expressing the short hairpin RNA (shRNA) against RNF20 exhibited slower kinetics in removing γH2AX following the treatment by the DNA-damaging agent VP-16(etoposide) (Fig. 7a, b), implying that RNF20 promotes the repair of DNA damage. Indeed, stable RNF20 depletion by shRNA impaired hRad51 foci formation upon VP-16 treatment and attenuated HR repair, as measured using immunostaining and the I-*Sce*I-induced DR-GFP reporter, respectively (Fig. 7c–e and Supplementary Fig. 13a)[62]. Consequently, RNF20 knockdown cells were susceptible to VP-16 (Fig. 7f and Supplementary Fig. 13b). As expected, all these defects were fully rescued by introducing a plasmid expressing the WT *RNF20* allele (Fig. 7a–f and Supplementary Fig. 13b). Notably, these defects can also be partially compensated by introducing the *RNF20-RD∆* mutant allele that lacks the ability to catalyze H2Bub (Fig. 7a–f and Supplementary Fig. 13b, c). These results suggest that RNF20 indeed exerts an E3 ligase-independent role in promoting hRad51 loading and HR repair.

As previously noted, RNF20 depletion also impaired BRCA1 foci formation (Supplementary Fig. 13d, e)[43,44]. However, introducing the *RNF20-RD∆* mutant allele did not obviously improve BRCA1 recruitment, suggesting that the ligase-independent role of RNF20 acts at downstream rather than upstream of the HR repair (Supplementary Fig. 13d, e). Interestingly, we noted that RNF20 and BRCA2 appear to display an additive effect in promoting HR repair, suggesting that RNF20 acts at least partially independent of BRCA2 in this process (Supplementary Fig. 14a, b). However, the role of RNF20 in HR repair is completely dependent on hRad51, since additional depletion of RNF20 in hRad51 knockdown cells did not further reduce the HR efficiency (Supplementary Fig. 14c, d).

## Discussion

A rate-limiting step during HR is the assembly of the Rad51-ssDNA filament. Efficient assembly of Rad51 filament requires surmounting the inhibition of excess ssDNA-bound RPA and the dismantling activities of anti-recombinases[10,26,27]. Our current understanding of how cells precisely modulate these steps to ensure proper HR is still limited, especially when encountering different types of DNA lesions. Here, we provide evidence that the evolutionarily conserved ubiquitin ligase Bre1 and its human homolog RNF20, a tumor suppressor, function as HR mediator proteins to regulate recombination in an E3 ligase-independent manner. We show that Bre1 and RNF20 exert multiple effects on regulating the stability or dynamics of Rad51 filaments: (1) stimulate Rad51 binding to ssDNA while restraining its loading to dsDNA; (2) accelerate the dynamics of Rad51 binding onto ssDNA; (3) stimulate the replacement of ssDNA-bound RPA by Rad51; (4) protect the Rad51-ssDNA filament by counteracting Srs2 or FBH1; (5) stimulate Rad51-mediated strand exchange reaction. Together with previous studies, we propose that Bre1/RNF20 plays dual roles in promoting HR repair. One is to catalyze the DNA damage-induced H2B ubiquitination that relaxes the closed chromatin structure at DSBs to allow access to repair machinery[43,44,47]. The other is to function as a recombination mediator, as described above (Supplementary Fig. 15a).

As an HR regulator, Bre1/RNF20 shares several common features with Rad52 or BRCA2. These proteins can interact with Rad51, direct Rad51 to ssDNA, and facilitate Rad51 nucleation on RPA-coated ssDNA. Meanwhile, they can stabilize the Rad51 filament and stimulate homologous pairing and strand exchange. As a result, their absence results in defective HR repair and reduced survival to DNA-damaging agents[12–17,19,63]. However, Bre1/RNF20 and BRCA2, but not yeast Rad52, can inhibit Rad51 loading onto dsDNA[13]. Our results

suggest that Bre1 acts additively with Rad52 or Rad57 in promoting HR repair. Similarly, RNF20 exhibits an additive effect with BRCA2 in stimulating HR repair. In contrast to the known mediators, Bre1/RNF20 did not exhibit a noticeable ssDNA binding ability, yet they can directly interact with RPA, Rad51, and Srs2/FBH1[48]. Thus, Bre1/RNF20 likely functions via a distinct mechanism from the known mediators in conquering the inhibitory effect of RPA and Srs2/FBH1. Interestingly, we found that Bre1 interacts with RPA, Rad51, and Srs2 via closely clustered motifs within a short-disordered region (Supplementary Fig. 15b)[48]. Although the RPA-interacting patch on Bre1 appears to be conserved across different species[48], the amino acid sequence of the Rad51- or Srs2-interacting patch on Bre1 is not conserved (Supplementary Fig. 15b). Thus, the RNF20 motif that mediates the interaction with hRad51 or FBH1 remains to be determined. How cells coordinate these interactions to finely regulate Rad51 filament dynamics and the recombination process remains to be explored.

An important function of Bre1/RNF20 is to protect the Rad51 filament from the disruption of Srs2 or FBH1. Interestingly, Bre1 can physically displace Srs2 from ssDNA without affecting its ATPase activity. We speculate that the binding of Bre1 to Srs2 might change Srs2 conformation, leading to its unloading from ssDNA. In yeast, the Rad55-Rad57 complex, Rad52, and the Shu complex were reported to counteract Srs2 activities, yet the biochemical evidence for Rad52 or Shu is lacking[30–32]. Multiple human proteins, including BLM, FBH1, PARI, REQL5, and FIGNL1, could disrupt hRad51 filament[33–37]. However, how their activities are restrained remains to be a critical question. The human hRad51 paralog SWSAP1 was shown to protect the hRad51 filament by antagonizing the anti-recombinase FIGNL1[35]. Here, we identified that RNF20 exhibited an ability to restrain the activity of FBH1.

A key question remaining elusive is why cells need multiple recombination mediator proteins. It was believed that Rad52 and BRCA2 are the primary mediators, while others appear to play subsidiary roles. One scenario is that these mediators cooperate to serve as a functional ensemble, as reflected by Rad52, Rad55-Rad57, and the Shu complex. Alternatively, some of these accessory factors may have specific roles in repairing certain types of DNA lesions. For example, disruption of the Bre1-Rad51 interaction specifically sensitizes cells to the DNA lesions induced by bleomycin or phleomycin. Thus, in addition to repair the HO-induced DSBs, Bre1/RNF20-mediated mechanism is probably particularly important for the recombination events associated with ssDNA gaps or DSBs related to apurinic/apyrimidinic sites or base loss that are common substrates generated by bleomycin-family drugs[49–51]. In support of this notion, the Shu complex was found to mediate recombination in DNA damage tolerance via binding the abasic sites at replication intermediates[64]. Finally, these accessory mediators may act in different chromatin contexts. Actively transcribed regions of a genome are known to be preferentially repaired by the accurate HR pathway, yet how this is achieved remains partially understood[65–67]. Since Bre1/RNF20-mediated H2B ubiquitination is linked with active transcription[38,39], it will be intriguing to examine whether Bre1/RNF20 plays a role in coupling transcription and HR repair or mediating RNA-templated HR repair[66,68,69].

In conclusion, we have identified the conserved ubiquitin ligase Bre1/RNF20 as an HR mediator that physically interacts with Rad51 to stimulate Rad51 nucleation and strand exchange while counteracting the activity of the Srs2 or FBH1 helicase. This report shows that a ubiquitin ligase can function as a recombination mediator. Advanced single-molecule and Cro-EM techniques could help to elucidate the detailed mechanisms by which Bre1/RNF20 regulates recombination. RNF20 is a tumor suppressor linked to several types of cancers[40]. Our study extends the view on how RNF20 may function to preserve genome stability and suppress tumorigenesis. Notably, the N-terminal

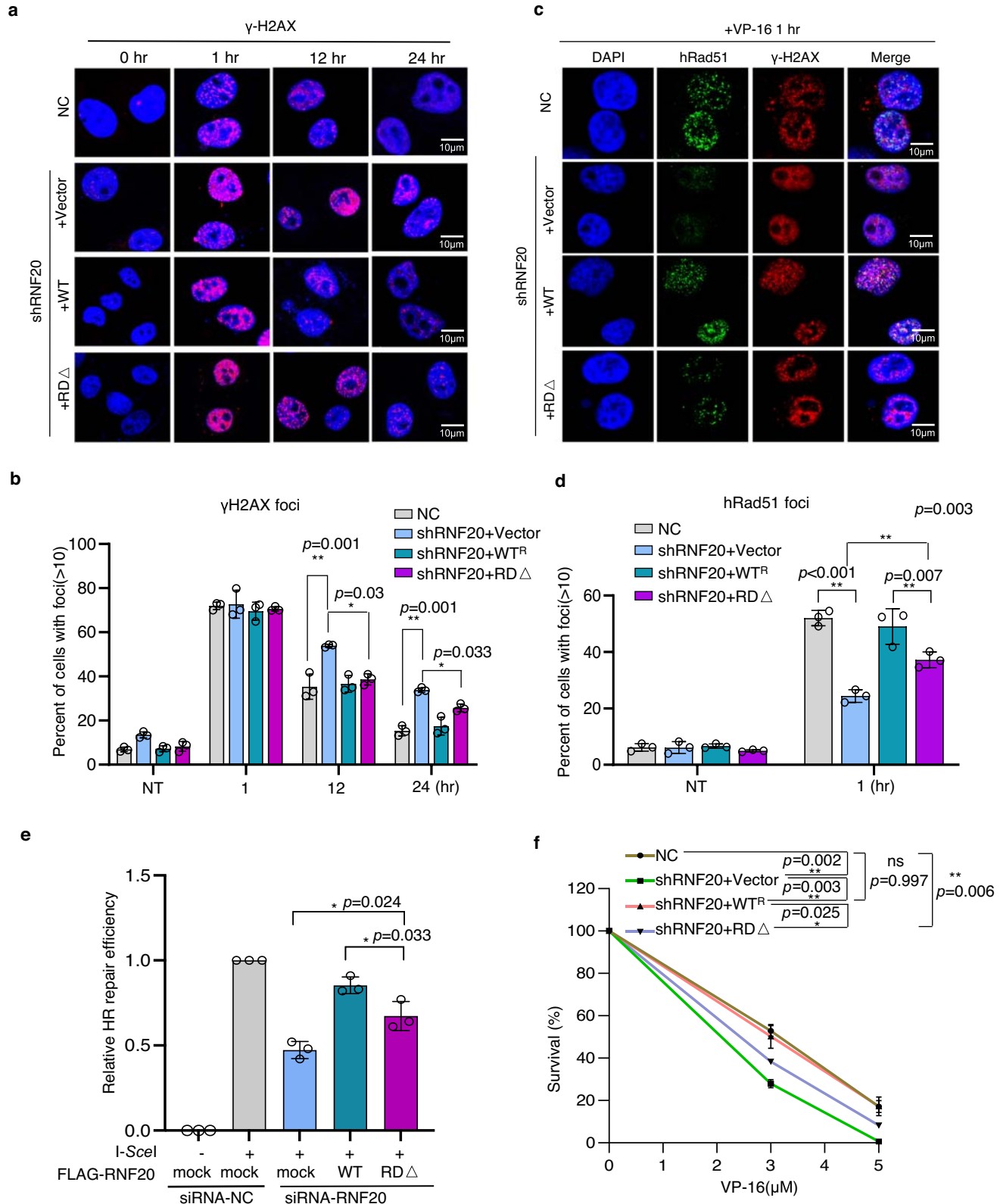

segment of RNF20 that interacts with hRad51 harbors two groups of clustered mutations in uterine corpus endometrial carcinoma patients (Supplementary Fig. 15c). It will be intriguing to investigate whether the occurrence of this cancer is relevant to an altered RNF20-hRad51 interaction. Our work could potentially provide alternative strategies to treat bleomycin-resistant cancer using inhibitors targeting RNF20 or its interaction with hRad51.

## Methods

### Yeast strains, human cell culture, and reagents

Yeast strains used in this study are presented in Supplementary Data 1. HEK293T (#CRL-3216) and Hela (#CCL-2) cells were obtained from ATCC and maintained in Dulbecco's modified Eagle's medium (DMEM) supplemented with 10% fetal bovine serum (FBS) and penicillin/streptomycin (P/S) at 37 °C with 5% $CO_2$. U2OS DR-GFP cell line was

**Fig. 7 | RNF20 can promote RAD51 loading and HR repair in a ligase-independent manner. a, b** Immunostaining and quantification of γH2AX foci number along with the recovery post VP-16 treatment in indicated Hela cells. The WT cells or RNF20 shRNA cells transfected with an empty vector or the plasmid expressing the full-length *RNF20* or *RNF20-RDΔ*allele were used for experiments. Cells were treated with VP-16(10 μM) for 1 h and then recovered in fresh media. Cells were collected at indicated time points and stained with DAPI or the antibody against γH2AX for immunofluorescence analysis. Data were the mean ± SD of three independent experiments (*n* = 3). **c, d** Immunostaining and quantification of

hRad51 foci formation in indicated Hela cells at 1 h after VP-16(10 μM) treatment. Cells were stained with DAPI and the antibody against hRad51 or γH2AX. Data were the mean ± SD of three independent experiments (*n* = 3). **e** Plot showing the HR efficiency measured with the DR-GFP reporter for indicated U2OS cell lines. Data are the mean ± SD of three independent experiments (*n* = 3). **f** Survival curve for the control or indicated Hela cells upon exposure to indicated doses of VP-16 treatment. Data were the mean ± SD of three independent experiments (*n* = 3). n.s. no significance; $*p < 0.05$, $**p < 0.01$, $***p < 0.001$ (Student *t*-test, two-tailed). Source data are provided as a Source Data file.

kindly provided by Dr. Xingzhi Xu (Shenzhen University). The sequences of PCR primers and siRNA or shRNA used in this study are listed in Supplementary Data 1.

## Proteomic sample preparation

Yeast-growing cells treated with 0.1% MMS (for 90 min) were collected and lysed on a bead beater in lysis buffer (100 mM HEPES, pH 8.0, 20 mM MgCl$_2$, 150 mM NaCl, 10% glycerol, 0.4% Nonidet P-40, 0.1 mM EDTA plus protease and phosphatase inhibitors). The extract was clarified by centrifugation at 12,000 × *g* for 10 min at 4 °C. The purified recombinant GST-Bre1 immobilized to the glutathione agarose beads was incubated with the above yeast cell lysates at 4 °C overnight with agitation to perform pull-down assays. The purified GST was used as a control. Subsequently, the beads were extensively washed with the lysis buffer containing 200 mM NaCl, and the proteins were eluted by boiling beads in 2xSDS loading buffer. Proteins were separated by an SDS-PAGE (10%) and digested by in-gel tryptic digestion. The gel pieces containing proteins were destained with 50 mM ammonium bicarbonate dissolved in 50% acetonitrile (1:1, vol/vol) and reduced by 10 mM DTT at 56 °C for 60 min, followed by alkylation with 55 mM iodoacetamide at room temperature for 45 min. Gel pieces were dehydrated in acetonitrile, rehydrated in 10 mM ammonium bicarbonate, and digested with trypsin (5 ng/μL, Promega) at 37 °C overnight. Peptides were extracted with 50% acetonitrile in 5% formic acid. The peptides were loaded on C18 stage tips twice and were desalted with 0.1%TFA and then dried using a speed vac to complete dryness. Mass spectrometry analysis was carried out at Wuhan University Proteomics Core Center. Each sample was analyzed once to produce a raw protein interactome list.

## Nano LC-MS/MS analysis

An analytical column (75 um inner diameter) packed with reversed-phase Repro-Sil Pur C18-AQ 1.9-um resin was prepared for sample loading. The dried samples were resuspended in 10 μL of LC-MS buffer A (0.1% FA) and then injected (3 μL) onto an Easy-nano LC system (Thermo) coupled online with Q Exactive HF mass spectrometer (Thermo Scientific). The HPLC gradient was set as follows: 2–5% Solvent B (0.1% FA in ACN) over 4 min, 5–35% B over 56 min, 35–44% B over 5 min, 44–90% B over 5 min, 90% B for 10 min at a flow rate of 200 mL/min. For data-dependent acquisition (DDA), full-MS (m/z 350–1600) were acquired in the orbitrap with a resolution of 60,000 and AGC target of le6. For MS/MS, the top 25 most abundant ions were automatically selected in each MS scan and fragmented in HCD mode at a resolution of 15,000 and AGC target of leS. Normalized collision energy (NCE) was set to 28. The LC-MS/MS data were processed and searched against the *Saccharomyces cerevisiae* Uniprot database using Proteome Discoverer (Version 2.5) for protein identification. A fully trypsin-specific search was chosen, and a maximum of two missed cleavages were allowed in the process. Fixed modification of carbamidomethyl at cysteine was chosen. The MS and MS/MS results were searched with a peptide ion mass tolerance of 10 ppm and a fragment ion mass tolerance of 0.02 Da. The raw protein interactome list generated in this study is provided in Supplementary Data 2. The interaction between Rad51 and Bre1 was verified by immunoprecipitation and pull-down assays.

## DNA damage sensitivity assay

Yeast cells were cultured in YEPD medium overnight. Undiluted or 1/10 serial diluted cell cultures were spotted onto YPD plates containing indicated concentrations of DNA-damaging agents. Plates were incubated at 30 °C for three days before taking pictures.

## Fluorescence microscopy

Cells were treated with phleomycin(20 μg/ml) for 2 h and then were released into a fresh YEPD medium without the drug for recovery. Rad52-YFP foci formation in live cells was examined using a ZEISS LSM 880 fluorescence confocal microscope carrying an Airyscan with a 63 x oil immersion objective lens and a YFP filter. Fluorescent images were captured and processed using ZEISS Blue Lite2 software.

## Immunostaining

Cells were cultured on coverslips and were treated with 10 μM VP-16 for the indicated times. Cells were then washed with PBS, fixed with 4% paraformaldehyde for 10 min at room temperature, and were subsequently permeabilized in 0.5% Triton X-100 solution for 10 min. After being blocked by 5% BSA, cells were incubated with primary antibodies overnight and subsequently secondary antibodies for 1 h. Coverslips were then mounted using DAPI containing anti-fade. Images were captured using a Leica SP8 Laser confocal optical imaging platform. Images were processed using Leica Application Suite X. The percentage of cells carrying indicated foci was calculated after analyzing three independent experiments. Approximately 150 cells were counted for each sample. Antibodies used for immunofluorescent staining are as follows: anti-gamma-H2AX (Abcam; Mouse; ab22551; 1:100 dilution), anti-Rad51(Abcam; Rabbit; ab133534; 1:400 dilution), anti-Brca1(Santa Cruz Biotechnology; Mouse; sc-6954; 1:100 dilution), and anti-FLAG (Cell Signaling Technology; Rabbit; #14793; 1:200 dilution). The following secondary antibodies were used: Alexa Fluor 488 Goat Anti-Mouse (Life Technologies; A-11008; 1:1000) and Alexa Fluor 594 Goat Anti-Rabbit (Life Technologies; A-11012; 1:1000).

## The survival rate of DSB repair by ectopic recombination

To measure cell viability from DSB repair by ectopic recombination, cells were cultured in the pre-induction medium (YEP-Raffinose) overnight to the log phase. Cells were then diluted and plated on YEPD and YEP-Gal plates, respectively, followed by incubating at 30 °C for 3 to 5 days. Survival rate = (Number of colonies grown on YP-Galactose)/(Number of colonies on YPD × dilution fold) × 100%. At least three independent tests were performed for each strain.

## Chromatin immunoprecipitation (ChIP)

ChIP and qPCR assays were carried out as previously described[70]. Log phase yeast cells ($-1 \times 10^7$ cells/ml) grown in YEP-Raffinose medium were subject to DSB induction by the addition of 2% galactose. Cells were harvested at 0 or 4hrs after DSB induction and lysed with glass beads in lysis buffer (50 mM HEPES pH 7.5, 140 mM NaCl, 1 mM EDTA, 0.1% NaDOC, 1% Triton X-100, and 1 mM PMSF, 1 mg/ml Bacitracin, 1 mM Benzamidine) supplemented with protease inhibitors. Chromatin DNA was sheared to an average size of ~250 bp using a Diagenode Bioruptor. IP samples were incubated with the anti-FLAG (Cell Signaling Technology; Rabbit; #14793; 1:200 dilution) or anti-H2B-K120-ub

(Cell Signaling Technology; Rabbit; #5546; 1:200 dilution) antibody overnight at 4 °C, followed by incubation with protein G-agarose beads (Cytiva, #17061805) for 4 h at 4 °C. The protein-bound beads were washed twice with lysis buffer, twice with lysis buffer containing 500 mM NaCl, twice with the wash buffer (10 mM Tris-HCl pH 8.0, 0.25 M LiCl, 1 mM EDTA, 0.5% NP-40 substitute and 0.5% NaDOC), and twice with 1x TE. Protein/DNA complexes were eluted with the elution buffer (10 mM Tris-HCl pH 8.0, 10 mM EDTA pH 8.0, 1%SDS) and incubated at 65 °C overnight to reverse crosslinking. Samples were digested with proteinase K at 37 °C for 6 h. DNA was purified by phenol extraction and ethanol precipitation. Purified DNA samples were analyzed by real-time quantitative PCR (StepOnePlus, 4376592, ABI) with primers that specifically anneal to DNA sequences located at indicated distances from the DSB using the following conditions: 95 °C for 10 min; 40 cycles of 95 °C for 15 s, 60 °C for 1 min, and 72 °C for 30 s.

## Expression of recombination protein and GST pull-down assay

The recombination proteins (6xHis-Bre1, 6xHis- EK2A, 6xHis- DK2A, 6xHis-Rad51, 6xHis-UBA1, 6xHis-Ub, 6xHis-Rad6, GST-Bre1, GST-Bre1-EK2A, GST-Rad51, GST-PPase, ScRPA, GST-Srs2, GST-FBH1, hRPA, GST-RNF20, GST-RNF20N, 6xHis-hRad51, and GST-hRad51) were expressed in *E.coli* BL21 (DE3). A single colony was inoculated into 1 L of LB medium containing 100 mg/ml ampicillin (or kanamycin) and induced at 0.8 OD600 using 0.5 or 1 mM IPTG. Cells were grown overnight at 16 °C and then harvested by centrifugation at $4000 \times g$ for 20 min at 4 °C and were frozen at −80 °C until use. The cell pellet was resuspended in lysis buffer (20 mM Tris-HCl, pH 7.4, 50 mM NaCl, 0. 5 mM EDTA, 10% glycerol and the protease inhibitor cocktail) and then lysed by sonication. The lysate was clarified by centrifugation at $12,000 \times g$ for 30 min at 4 °C. All the proteins were concentrated and stored at −80 °C.

For the GST pull-down assay, recombinant GST-Bre1 (or GST-Bre1-EK2A) or GST-RNF20 was immobilized on 30 μL of bed volume of glutathione agarose beads (ABclonal). After washing with lysis buffer, the resin was then incubated with 6xHis-tagged Rad51 or 6xHis-tagged FBH1 proteins at 4 °C for 4 h on a rotator. The beads were washed extensively with wash buffer (20 mM Tris-HCl, pH 7.4, 200 mM NaCl, 0. 5 mM EDTA, 10% glycerol), and bound proteins were detected by Western blot or Coomassie brilliant blue staining of SDS-PAGE gels.

For the Bre1-Srs2 interaction assay, the recombinant GST-Srs2 was immobilized on 30 μl of bed volume of glutathione agarose beads for 1 h at 4 °C with agitation. After washing with lysis buffer, the resin was then incubated with 6xHis-Bre1(or 6xHis-Bre1-DK2A) protein at 4 °C for 4 h on a rotator. The beads were washed extensively with wash buffer (20 mM Tris-HCl, pH 7.4, 200 mM NaCl, 0. 5 mM EDTA, 10% glycerol), and bound proteins were separated by SDS-PAGE gels followed by Coomassie brilliant blue staining.

## Protein purification

To purify GST-tagged proteins, cell lysates were centrifuged at $15,000 \times g$ for 20 min and the supernatant was applied to GST affinity resin (ABclonal). The resin was equilibrated with the binding buffer (140 mM NaCl, 2.7 mM KCl, 10 mM Na₂HPO4, 1.8 mM KH₂PO4, pH 7.4). After equilibration, the resin was then incubated with the supernatant for 1 h on a rotator at 4 °C. Beads were washed eight times with 10 ml of GST wash buffer (50 mM Tris-HCl pH 8.0, 100 mM NaCl, 5 mM DTT, 1 mM EDTA, 0.5%Triton X-100). The bound protein was eluted by the elution buffer containing 50 mM Tris-HCl, 20 mM reduced glutathione pH 8.0, and the protease inhibitor cocktail. The fractions were then concentrated using a 30-kDa cut-off super-filter tube (Millipore) and stored at −80 °C for further use.

To release Rad51 from the GST tag, purified GST-Rad51 proteins were incubated with pre-equilibrated glutathione agarose in a binding buffer (20 mM Tris-HCl, pH 7.4, 200 mM NaCl, 0. 5 mM EDTA, 10% glycerol, and the protease cocktails) for 1 h on a rotator at 4 °C. The

beads were washed three times with wash buffer that contains 50 mM Tris-HCl pH 8.0, 1 M NaCl, 1 mM DTT, 0.5 mM EDTA, 0.1% Triton, and the protease cocktail. After washing, a 2/3 column volume of cleavage buffer (50 mM Tris-HCl pH 8.0, 2 M NaCl, 1 mM DTT, 40 μL 2 mg/mL PPase, protease inhibitors) was added to the resin and incubated overnight on a rotator at 4 °C. Rad51 without GST tag was collected and concentrated using a 30-kDa cut-off super-filter tube (Millipore) and stored at −80 °C.

For purification of His-tagged associated proteins, the cell lysate was centrifuged at $15,000 \times g$ for 20 min and the supernatant was mixed with Ni-NTA agarose beads (ABclonal) pre-equilibrated in buffer containing 20 mM Tris-HCl, pH 7.4, 100 mM NaCl, 0. 5 mM EDTA, 10% glycerol, 20 mM imidazole and 1 mM DTT for 2 h on a rotator at 4 °C. Protein was eluted by a step gradient of imidazole (50, 100, 200, 300, 400, and 500 mM) in the buffer described above. The eluent was then concentrated using a 30-kDa cut-off super-filter tube (Millipore) and stored at −80 °C.

For expression of the yeast or human RPA complex, the DNA sequence of RPA70/Rfa1 was cloned into a pFastbac1 vector engineered with a cleavable N-terminal GST tag, and the DNA sequence of RPA32/Rfa2 and RPA14/Rfa3 were cloned into a modified pFastbac_dual vector (Invitrogen). Baculoviruses were generated with Sf9 insect cells using the Bac-to-Bac system (Invitrogen). The RPA heterotrimer was expressed by co-infecting Hi-5 insect cells with both viruses. Forty hours after infection, cells were collected by centrifugation at $1000 \times g$ for 15 min and lysed in a buffer containing 20 mM HEPES buffer pH 7.0, 250 mM NaCl, 10% glycerol, 0.3 mM TCEP, 1 mM phenylmethylsulfonyl fluoride (PMSF). The cell lysate was centrifuged at $30,000 \times g$ for 1 h, and the supernatant was incubated with 5 ml of GST-agarose beads for 1 h at 4 °C. The protein with GST tag was eluted with elution buffer (lysis buffer plus 200 mM glutathione). The GST tag was subsequently cleaved by GST-TEV protease. After cleavage of the GST tag, the protein was further purified on a Heparin column and a Superdex 200 increase gel filtration column (Cytiva). Purified RPA heterotrimer was concentrated in 20 mM Tris-HCl, 200 mM NaCl, 0.3 mM TCEP, pH 8.0, with 15% glycerol for storage.

## Immunoprecipitation (IP) and Western blotting

Yeast cells culture (A600 ∼ 1.0) with or without MMS treatment were collected and lysed on a bead beater in lysis buffer (100 mM HEPES, pH 8.0, 20 mM MgCl₂, 150 mM NaCl, 10% glycerol, 0.4% Nonidet P-40, 0.1 mM EDTA plus protease and phosphatase inhibitors). The extract was clarified by centrifugation at $12,000 \times g$ for 10 min at 4 °C, followed by incubating with protein G-agarose beads for 1 h at 4 °C to preclear non-specific binding. After centrifugation, the supernatant was incubated with anti-HA (MBL) or anti-FLAG (MBL) antibodies at 4 °C overnight with agitation. Protein G-agarose beads were added and the mixtures were incubated for another 3 h at 4 °C. Subsequently, the beads were subjected to extensive washing at 4 °C with the lysis buffer. Immunoprecipitated proteins were eluted by boiling beads in 2xSDS loading buffer for 5 min.

To detect the interaction between RNF20 and hRad51 or FBH1, HEK293T cells were co-transfected with FLAG-tagged and HA-tagged constructs. Cells were then lysed with NETN buffer (20 mM Tris-HCl, pH 8.0, 100 mM NaCl, 1 mM EDTA, and 0.5% Nonidet P-40) containing protease inhibitors (1 μg/mL of aprotinin and leupeptin) on ice for 15 min. The cell lysates were centrifuged at $12,000 \times g$ at 4 °C for 5 min, and the resulting supernatants were incubated with protein G-sepharose coupled with 3 μg of anti-FLAG antibody for 16 h at 4 °C with gentle rocking. The bead-bound proteins were washed three times with NETN buffer and then boiling beads in 2xSDS loading buffer for 5 min.

Samples were resolved on an 8 or 12% SDS-PAGE gel and transferred onto a PVDF membrane (Immobilon-P; Millipore) using a semi-dry method. Primary antibodies used for immunoblot analysis are as

follows: anti-FLAG (Cell Signaling Technology; Rabbit; #14793; 1:1000 dilution), anti-H2Bub(Cell Signaling Technology; Rabbit; #5546; 1:1000 dilution), anti-RNF20(Abcam; Rabbit; ab32629; 1:3000 dilution), anti-FBH1(Abcam; Mouse; ab58881; 1:1000 dilution), anti-Rad51(Abcam; Rabbit; ab133534; 1:3000 dilution), anti-Brca2 (ABclonal; Rabbit; A2435; 1:1000 dilution), anti-HA (ABclonal; Mouse; AE008; 1:5000 dilution), anti-H2B (ABclonal; Rabbit; A1958;1:5000 dilution), anti-GAPDH (ABclonal; Mouse; AC002,1:5000 dilution), anti-His (Proteintech; Mouse;66005-1-Ig; 1:5000 dilution), and anti-GST (Proteintech; Mouse;66001-2-Ig; 1:5000 dilution). Horseradish peroxidase (HRP) conjugated secondary antibodies used are HRP goat anti-mouse IgG (Jackson Immuno #115-035-003, 1:10000) and HRP goat anti-rabbit IgG (Jackson Immuno #111-035-003, 1:10000). Blots were developed using the Western Blotting substrate (Bio-Rad).

### NCP ubiquitination assay

NCP preparation and in vitro ubiquitination assay were performed as described previously in ref. 48.

The expression and purification of His-tagged proteins, including 6×His-UBA1, 6×His-Rad6, GST-Ub, 6×His-Bre1, and 6×His-Bre1-*EK2A* were carried out as described above. NCP was prepared from *bre1Δ FLAG-H2B* yeast cells. To prepare NCP, yeast cells were grown at 30 °C in YEPD to an $OD_{600}$ of 0.8. The spheroplasts were prepared by incubating cells in the digestion buffer (1 M sorbitol, 10 mM β-ME, 0.5 mg/mL Zymolyase 100 T, 50 mM Tris-HCl, pH = 7.5) at 30 °C for about 20 min. The spheroplasts were treated with 0.25 U/μL of micrococcal nuclease in the reaction buffer (0.5 mM spermidine, 1 mM β-ME, 0.075% NP-40, 50 mM NaCl, 5 mM MgCl₂, 5 mM CaCl₂, 10 mM Tris-HCl, pH = 8.0) at 37 °C to prepare nucleosomes. Reactions were stopped after 15 min by adding 20 mM EDTA and 1% SDS. For in vitro ubiquitination assays, 200 nM UBA1, 72 μM ubiquitin, 6 μM Rad6, 36 μM Bre1 or bre1-EK2A, 4 mM ATP, 0.1 mM DTT, and 5 μM NCP were added to the reaction (50 mM Tris-HCl, pH 8.0, 50 mM NaCl, 50 mM KCl, and 10 mM MgCl₂) with a total volume of 80 μL. The reaction was incubated at 37 °C for 90 min and stopped by adding 5×SDS loading buffer. The products were resolved on a 15% SDS-PAGE followed by Western blot analysis with anti-FLAG (F3165; Sigma; 1:3000 dilution).

### Streptavidin pull-down assay

A 5′-biotinylated 90-nt oligonucleotide (CGACAGGTCATGGCCGTACATGATATCCTCGAGCGTCCTGTTGCAACTTACACTCTGAATAGCCG AATTCTTAGGGTTAGGGTTAACA) was immobilized to 30 μl of streptavidin MagBeads (GenScript) in TES buffer (10 mM Tris, 1 mM EDTA, 2 M NaCl, pH 7.5) at room temperature for 30 min. After an extensive wash with 1xPBS supplemented with 1 mM EDTA, 5 μM of purified yeast or human Rad51 was added to the reaction in the binding buffer (42 mM Tris-HCl [pH 7.5], 15 mM KCl, 90 mM NaCl, 2 mM ATP, 1 mM EDTA, 0.5% NP-40, 1 mM DTT, 20 μg/mL BSA, and 5% glycerol). The mixture was incubated at 37 °C for 30 min, followed by washing once with the binding buffer. An increasing amount of purified 6xHis-Bre1 or 6xHis-bre1-EK2A protein was subsequently added to the reaction. The mixture was incubated for 15 min at 37 °C, and the beads were washed three times with the binding buffer and twice with the binding buffer without BSA. Bound protein was eluted by boiling in 2xSDS loading buffer and detected by Western blot or Coomassie brilliant blue staining.

To detect the replacement of RPA by Rad51, biotinylated oligonucleotide was premixed with yeast or human RPA with indicated concentrations in a 20 μl volume in the binding buffer and incubated for 10 min at 37 °C. Then, purified yeast or human Rad51 (0.6 μM) was added either with or without Bre1(RNF20) or bre1-EK2A (1.2 μM) to the reaction and was further incubated for 15 min at 37 °C. Next, pre-equilibrated streptavidin MagBeads (30 μl) were added to the reaction and incubated at 4 °C for another 30 min with agitation. Then, the beads were washed and eluted as above. Eluted proteins were detected

by Western blot with an anti-His monoclonal antibody (ABclonal; Mouse; AE003; 1:10,000 dilution).

For the competition pull-down assay, purified yeast Rad51(5 μM) was preincubated with a gradient amount of Bre1(1–5 μM) in a 10 μl volume in the binding buffer (42 mM Tris-HCl, pH 7.5, 15 mM KCl, 90 mM NaCl, 1 mM EDTA, 0.5% NP-40, 1 mM DTT, 20 μg/mL BSA, and 5% glycerol) for 10 min at 37 °C (Rad51 alone was used as a control). Then, a biotinylated oligonucleotide (ssDNA) (200 nM) was added to the reaction together with or without an equal amount of homologous dsDNA and incubated for another 10 min. Next, pre-equilibrated streptavidin MagBeads (30 μl) were added to the reaction and incubated at 4 °C for 30 min with gentle rocking. Subsequently, the beads were washed and eluted. Eluted proteins were detected using Western blot with antibodies.

### Electrophoretic mobility shift assay (EMSA)

An increasing amount of yeast Rad51 or hRad51 was incubated with circular M13 mp18 ssDNA(100 nM) in the buffer containing 42 mM MOPS (pH 7.2), 3 mM MgCl₂, 1 mM DTT, 20 mM KCl, 25 μg/ml BSA, and 2.5 mM ATP in a final volume of 20 μl at 37 °C for 10 min. To detect whether Bre1 or RNF20N stimulates Rad51-ssDNA assembly or co-complexes with the filament, a stoichiometric amount of Bre1, bre1-EK2A, or RNF20N was incubated with the Rad51-ssDNA filament in the reaction buffer in a final volume of 20 μl at 37 °C for 15 min. To test the effect of Bre1 on Srs2 binding on ssDNA, Srs2 was preincubated with 100 nM M13 mp18 ssDNA or the Y-shape or 3′-flap DNA in the reaction buffer in a final volume of 15 μl at 37 °C for 10 min. Subsequently, Bre1(or Rad52) was added to the reaction and incubated for another 10 min. 0.25% of glutaraldehyde was then added to the mixture and the reaction was incubated for 15 min at room temperature to crosslink the protein-DNA complexes. The complexes were separated by a 0.8% agarose gel and stained with GelRed. Signals were detected through the gel imaging system. Band intensities were quantified with Image J.

### DNA strand exchange reaction

The 80-nt Oligo1 (100 nM) was first coated with purified yeast or human RPA (150 nM) in the reaction buffer containing 50 mM Tris-HCl, pH 7.5, 2 mM MgCl₂, 2 mM ATP, 35 mM KCl, 1 mM dithiothreitol (DTT), and 100 μg/ml BSA in a final volume of 10 μl at 37 °C for 10 min. Following this, purified yeast or human Rad51(1.6 μM) together with Bre1 (RNF20N) or mutated Bre1-EK2A (0.8 or 1.6 μM) were then added to the reaction and incubated at 37 °C for 10 min. To test the effect of Srs2 or FBH1 on strand exchange, 0.4 μM of Srs2 or FBH1 was added to the reaction together with Rad51. The strand exchange reaction was initiated by adding 50 nM of a Cy3′-labeled 40-nt dsDNA. After incubation at 37 °C for 30 min, samples were deproteinized by the addition of 5 μl of 1% SDS solution containing proteinase K (1 mg/ml) at 37 °C for 10 min and analyzed by electrophoresis (10% polyacrylamide gel in 1xTBE buffer). Signals were imaged with a ChemiDOC MP scanner (Bio-RAD) and quantified with the Image J software. DNA substrates for DNA strand exchange:

Oligo1: 5′-TTATGTTCATTTTTTATATCCTTTACTTTATTTTCTCT GTTTATTCATTTACTTATTTTGTATTATCCTTATCTTATTTA-3′.Oligo2: Cy3-5′TAATACAAAATAAGTAAATGAATAAACAGAGAAAATAAAG-3′

Oligo3: 5′-CTTTATTTTCTCTGTTTATTCATTTACTTATTTTGTAT TA-3′

The dsDNA substrate was obtained by annealing the oligo2 and oligo3. The Cy3′ labeled duplex substrate was purified from a 10% polyacrylamide gel.

### Measurement of Srs2/Rad51 ATPase activity

ATP hydrolysis was determined using the ATPase/GTPase Activity Assay Kit (Sigma, Cat.No.MAK113) following the manufacturer's instructions. The reaction was carried out in the ATPase buffer containing 20 mM Tris-HCl, pH 7.5, 2.5 mM MgCl₂, 150 mM KCl, 5 mM

CaCl$_2$, 4 mM ATP, 1 mM DTT, and 100 μg/ml BSA at 37 ˚C. To measure the ATPase activity of Srs2, 1.5 μM purified Srs2 was incubated with or without 100 nM M13 mp18 ssDNA for 5 min at 37 ˚C. About 1.5 μM Bre1 was included in the reaction to test the effect of Bre1 on Srs2 ATPase activity. To determine the effect of Bre1 or RNF20 on Rad51 ATPase activity, 0.5 μM yeast or human Rad51 was incubated with or without 100 nM M13 mp18 ssDNA for 5 min at 37 ˚C, followed by incubating with Bre1 or RNF20 for another 10 min at 37 ˚C. To measure the level of ATP hydrolysis, a standard curve was generated by plotting the optical density (OD 620 nm) values of the standards against their concentrations. The ATPase activity was calculated according to the following formula: Enzyme activity (U/L) = $Pi$ (μ$M$)×40 μL/$S_V$ × $T$, where 40 μL = reaction volume, $S_V$ = sample volume (μL) added to well, T = reaction time (minutes).

### Single-molecule study
The 12.5 k-nt ssDNA was generated by one-sided PCR, and its two ends were labeled with digoxigenin and biotin groups, respectively. In MT experiments, the digoxigenin-labeled end of a single ssDNA molecule was anchored to the anti-digoxigenin-coated glass surface in a flow cell. Then, a superparamagnetic microbead (M-270, Dynal beads) was attached to the biotin-labeled end of the anchored ssDNA molecule. A pair of permanent magnets were used to attract the microbead and thus exert a constant force on the anchored ssDNA molecule. The extension of ssDNA was determined to be the separation between the microbead and glass surface. The assembling buffer contained 100 mM NaAc, 10 mM MgAc$_2$, 1 mM ATP, and 25 mM Tris-Ac pH 7.5. All experiments were performed at a constant force of 8 pN at 20 ℃. Protein concentrations used are indicated in the figure legends.

### Statistics and reproducibility
All statistical analysis was performed using GraphPad Prism (v.8.0). Statistical analysis was performed using two-tailed Student's $t$-tests unless mentioned otherwise. $p < 0.05$ was considered significant. The number of samples for each experiment is indicated in the figures or figure legends. For Western blot, pull down, or EMSA assays, typically, three independent experiments were carried out.

### Reporting summary
Further information on research design is available in the Nature Portfolio Reporting Summary linked to this article.

## Data availability
The mass spectrometry raw data are not available anymore. The list of interacting proteins identified by mass spectrometry in this study is provided in Supplementary Data 2. All remaining data supporting the findings of this study are available within the article and its supplementary information files. Source data are provided with this paper.

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

## Acknowledgements

We thank Dr. Xiangwei He (Zhejiang University) for the critical reading of the manuscript. This research was supported by grants from the National Natural Science Foundation of China [32070573 and 31872808], the National Key Research and Development Program of China [2021YFA1100503], the Fundamental Research Funds for the Central Universities [2042023kf0232], TaiKang Center for Life and Medical Sciences and Wuhan University Advanced Genetics Course Program to X.C.

## Author contributions

G.L. and J.L. performed most of the in vivo and in vitro experiments. B.H. performed some in vitro biochemical experiments. J.Y., J.Z., X.W., J.X., X.G., S.Z., and X.L. constructed plasmids, yeast strains, and purified some proteins. XH.Z. and XC.Z. designed and performed single-molecule studies. C.Z. and Y.W. supervised or carried out the purification of RPA and Rad51. G.L., J.L., and X.C. designed most of the experiments. X.C. supervised the work. X.C. and G.L. wrote the manuscript.

## Competing interests

The authors declare no competing interests.
