## [Peer Review File · Nature Communications]

Bre1/RNF20 promotes Rad51-mediated strand exchange and antagonizes the Srs2/FBH1 helicasesREVIEWER COMMENTS

Reviewer #1 (Remarks to the Author):

The manuscript by Liu et al identifies a novel function for Bre1 in promoting RAD51-driven recombination. It is presented that Bre1 binds Rad51, stimulates Rad51 ssDNA binding, inhibits dsDNA binding, promotes RAD51 strand exchange. Bre1 also interacts with Srs2, and promotes recombination additionally by preventing the action of a negative recombination regulator, Srs2. Bre1 thus appears to promote recombination by pretty much every mechanism ascribed to traditional recombination mediators to date. In the last part, a similar interplay is described for human proteins RNF20 and FBH1.

The topic of the study, regulation of recombination, is important and relevant. At first glance, I was impressed by the almost incredible amount of data, which exceed what would be normally included in several papers. Diving more deeper in the data, I noted a high number of inconsistencies.

Specific comments:

- Figure 1: Is the interaction patch in Bre1 conserved? Please present an alignment of this region.
- Figure 2g: please make the legend based on the color of the bars and not on the individual datapoints.
- Figure 2d: Facilitation of the RAD51 filament on ssDNA may shift bound DNA into wells by RAD51 alone; the experiment does not prove that Bre1 is a part of the complex.
- The authors argue that Bre1 remains a part of the complex once RAD51 filament assembles on DNA. It is somewhat difficult to imagine that such a filament would be promoted for strand exchange: it is known that RAD51 filament needs to be uninterrupted to facilitate strand exchange
- Figure 3h: the authors observed the replacement of RAD51 on RPA-coated ssDNA by Bre1. While the increase of RAD51 is apparent, there does not appear to be a corresponding decrease of RPA. Why is that?
- Figure 3h: Preincubation of RPA before RAD51 is added is expected to decrease RAD51 bound to ssDNA. Yet, no decrease is apparent: comparing lanes 4 and 7. Why is that?
- Assay in Fig. 3l: combination of 7.7 kbp ssDNA and 2.2 kbp dsDNA cannot give rise to nicked DNA after strand exchange.
- Furthermore, in panel 3m, stimulation of strand exchange is expected to enhance the utilization of dsDNA, i.e. the intensity of the dsDNA band should go down when Bre1 is present, which is not at all the case. While I do see increase in intensity of the strand exchange products, it does not make sense without the corresponding decrease of the DNA substrates. Additionally, ssDNA is already fully engaged even without Bre1.
- Figure 3d vs SI 6a: In panel 3a, 5 uM of RAD51 did not cause a shift of 100 nM of ssDNA to wells (in fact, only minimal DNA binding was observed). Yet, in panel SI 6a, already 2.5 uM of RAD51 shifted 100 nM dsDNA almost completely to wells. Why is that? The affinities are thought to be similar. Were the substrates of the same length?
- In panel 4d, the "nc" form (which is probably not nc, see above) runs below ssDNA, while in panel 3m, it runs above. Why is that?
- Panel 4d: I am wondering what happened to ssDNA in the presence of Srs2. As above, while I see that Bre1 increases the forms labeled as jm and nc, the proportion of the bands in the experiment does not make a lot of sense.
- The authors conclude that Bre1-Rad51 interaction is required for Bre1 to displace Rad51. However, displacement of Srs2 by Bre1 was observed even in the absence of Rad51 (Fig. 4h). How can the displacement occur when Bre1 and Srs2 do not directly interact without DNA (as noted)?
- Figure 5b: The experiment is a GST pulldown, what is pulled is GST-tagged Srs2. I am confused why there is more Srs2 pulled down in the presence of DNA. Do the authors suggest that pulling on DNA-bound Srs2 may bring other, DNA-bound Srs2 molecules bound to other locations on DNA? Along

these lines, Bre1 in the same experiment is bound to Srs2, or independently to other DNA sites? Increasing Bre1 concentrations (comparing lanes 5 and 6) brings down more Bre1 than Srs2, arguing that Bre1 might bind DNA independently.

- Please note whether the interaction patch between Srs2 and Bre1 is conserved (alignment of the interaction motif).
- Fig. 6i: similar concerns as noted above - the total content of DNA in the various lanes does not add up.

Reviewer #2 (Remarks to the Author):

Homologous recombination (HR) is one of the major double-strand break repair pathways. As the key recombinase, Rad51, once loaded onto single-stranded DNA generated from double-strand break processing, catalyzes homology search and strand exchange. In cells, mediator proteins, such as Rad52 and BRCA2, are required for Rad51 loading due to the RPA barrier. A key aspect of HR research centered upon the assembling and disassembling of the Rad51 nucleoprotein complex. In this manuscript by Liu et al, Bre1, previously known as an E3 ligase functioning in DNA repair, was found to directly interact with both Rad51 and Srs2, a 3'-5' helicase that possesses the anti-recombinase activity. With the separation-of-function mutants isolated that specifically disrupt the identified novel interface, the authors proceeded to apply biochemical and single molecule approaches and conclude Bre1 may function as a new mediator protein to facilitate Rad51 nucleoprotein filament assembling and to antagonizing the anti-recombinase activity from Srs2. Overall, the data on characterizing the protein-protein interface is solid and the novel functions of Bre1 in assisting Rad51 and antagonizing Srs2 are very interesting. The biochemical data on assessing the mediator activity of Bre1 are, however, less convincing, which, in conjunction with the limited experimental details included in the figure legends makes it very difficult to evaluate the manuscript in its current form. There are a few suggestions depicted below that will help further strengthen the manuscript.

Major points:

1. While some experimental details were provided in the method section, the figure legends were oversimplified. For quite a few experiments, it is not clear what type of substrates were used and at what concentration.
2. The key function of mediator proteins is to assist Rad51-catalyzed strand exchange reaction in the presence of RPA, which competes with Rad51 on ssDNA. In the strand exchange assays presented in Fig 3m, n, and Fig 4i, k, as indicated, 100 nM M13 substrate DNA, 250 nM of Rad51, and 100-170 nM RPA were used. Accordingly, there were on average 2-3 Rad51 molecules and 1-2 RPA molecules per each substrate DNA that is over 7000 nt in length. Under such conditions, both Rad51 and RPA were far from saturating the DNA. With such a concentration of RPA, its barrier effect was probably overlooked, which renders the conclusion of Bre1 as a mediator protein an over-interpretation of the data presented.
3. ATP binding and hydrolysis by Rad51 regulate its DNA binding. While ATP was included in the EMSA and Strand Exchange assays, it was not present in the streptavidin pull-down assay based on the buffer conditions provided in the method section.
4. Based on the model in the manuscript, a prediction would be the bre1-EK2A and bre1-DK2A mutants may impact the recombination rate. The authors do have a physical assay system in the manuscript (Fig 2d, e), it will be nice to compare the HR rate (crossover vs. noncrossover) in the wild-type, bre1-EK2A, and bre1-DK2A, which will largely strengthen the paper.
5. In Fig S1, bre1-6A mutation caused an increase in RPA loading, which indirectly affected Rad51

loading. Hence, for the bre1-EK2A mutant, a key control experiment that was missing is to examine its impact on RPA loading.

More specific points:

1. In Fig 3b, the molecular weight marker is missing, and it is not clear what percentage of gel was used. Assuming it is a western blot, what antibodies were used to probe?
2. Fig 3 legend states that 50 pM of 90 mer ssDNA was used in the streptavidin pull-down assays, but Fig 3b marked ssDNA as 100 nM.
3. In Fig 3h, the loading of RPA didn't seem to affect Rad51 binding, why?
4. In Fig 3m, a control reaction without ATP should be included. Also, the M13 single-strand DNA completely disappeared while only 40% reacted in lane 4, why?
5. In Fig 4g, the result will be clearer if a short ssDNA substrate was used to monitor Srs2 binding without/with Bre1 in the EMSA assay without cross-linking.

Reviewer #3 (Remarks to the Author):

This article reported a novel function of Bre/RNF20 in HR repair process via regulating the loading of Rad51 to ssDNA. The authors utilized genetic modified yeast strains to prove their hypothesis, and validated some of their results in human cell lines. It is potentially interesting for the field as the findings extend our understanding of how Bre/RNF20 and Rad51 work together in HR repair regulation. Although they discussed possible mechanisms, however, this manuscript lacks explanation of how Bre/RNF20 regulates the loading and unloading of proteins such as Rad51, Srs2 and RPA, since Bre/RNF20 itself has no binding to ssDNA. Moreover, most of the conclusions are based on either in vitro assay or overexpression, especially the results in human cells. For instance, the interaction assay for Rad51/RNF20 and FBH1/RNF20. In addition, experiment such as immunostaining or pull-down assay with genetic modified (KO or KD) cell lysates should be performed to show the loading or unloading of related proteins to DSB in vivo. Furthermore, whether the effects of Bre/RNF20 on HR repair are mediated by Rad51 should be clarified (using Rad51 KO or KD cells to detect the effects of RNF20 on HR).

Minor points:

Why the two amino acids are only important for some proteins (RPA, but not Srs2) unloading. What could be the possible explanation?

The authors claimed that Bre promotes the replacement of ssDNA-bound RPA by Rad51, but why the amount of PRA pull-down by ssDNA was not changed in lane 7 and lane 8 of Fig.3g? The same issue was observed in Fig.7f, where no decrease of ssDNA-bound RPA with the addition of RNF20N.

The total amounts of ssDNA and dsDNA in Fig3m should be the same, but why the lane 5,6 have more dsDNA left, although they have more jm and nc in the meantime.

How the authors did the normalization in Fig.3o? the amount of jm for sure is not increased with condition of pep104 in 500 nM.

What is the difference of lane 6,7 and lane 4, 5 of Fig.S6a? How the authors make the conclusion that Bre1 could not remove Rad51 from the pre-formed Rad51-dsDNA complex?

Will dsDNA mediate the interaction of Srs2 and Bre1 (Fig.5b)?

According to the text, it should be siRNA instead of shRNA in Fig.7a,7c.

Other issues: Manuscript needs to be revised carefully to avoid errors: Streptavidin instead of Streptavadin in Fig.4a; "p" or "p".

A point-by-point response to the reviewer's questions

We thank the Editor and all reviewers for their positive comments and constructive suggestions on the manuscript. We carefully performed many additional experiments as reviewers suggested to address their concerns. Below is our response to the questions raised by reviewers. The changes in Main text are marked in red.

REVIEWER COMMENTS

Reviewer #1 (Remarks to the Author):

The manuscript by Liu et al identifies a novel function for Bre1 in promoting RAD51-driven recombination. It is presented that Bre1 binds Rad51, stimulates Rad51 ssDNA binding, inhibits dsDNA binding, promotes RAD51 strand exchange. Bre1 also interacts with Srs2, and promotes recombination additionally by preventing the action of a negative recombination regulator, Srs2. Bre1 thus appears to promote recombination by pretty much every mechanism ascribed to traditional recombination mediators to date. In the last part, a similar interplay is described for human proteins RNF20 and FBH1.

The topic of the study, regulation of recombination, is important and relevant. At first glance, I was impressed by the almost incredible amount of data, which exceed what would be normally included in several papers. Diving more deeper in the data, I noted a high number of inconsistencies.

Answer: We appreciate the reviewer's recognition of the significance and efforts of our work. We carefully performed additional experiments to address these questions or concerns.

Specific comments:

- Figure 1: Is the interaction patch in Bre1 conserved? Please present an alignment of this region.

Answer: We aligned the amino acid sequences of the Rad51- and Srs2-interacting patches on Bre1 against Bre1 homologs from different species. We noted that the E500 K502 and D484 K486 motifs are not well conserved across species. Thus, the RNF20 motif that mediates the interaction with hRad51 or FBH1 remains to be determined. This result is incorporated in the manuscript (Main text, page 15, paragraph 1, lines 10-13; Extended Data Fig. 15b).

- Figure 2g: please make the legend based on the color of the bars and not on the individual datapoints.

Answer: It is now fixed (**Fig. 2g**).

- Figure 3d: Facilitation of the RAD51 filament on ssDNA may shift bound DNA into wells by

RAD51 alone; the experiment does not prove that Bre1 is a part of the complex.

Answer: In the ssDNA pull-down assay presented in **Fig. 3b**, we noted that Bre1 alone does not bind ssDNA or beads (Fig.3b, lanes 2-3), yet it can be captured in ssDNA pull down when Rad51 is present (lanes 4-6). Based on this result, we believe that Bre1 can associate with the Rad51-ssDNA filament. However, in case of this interaction is dynamic or unstable, we changed the word “co-complex” to “associate or interact” throughout the text to tune down the statement (Main text, Page 6, Paragraph 3, lines 1, 8 and 9).

- The authors argue that Bre1 remains a part of the complex once RAD51 filament assembles on DNA. It is somewhat difficult to imagine that such a filament would be promoted for strand exchange: it is known that RAD51 filament needs to be uninterrupted to facilitate strand exchange.

Answer: Previous studies showed that the recombination mediator Rad55-Rad57 associates with the Rad51-ssDNA complex to form a co-filament, and this is required to stabilize the Rad51-ssDNA filament and antagonize the disruption by Srs2 (**Liu et al, 2011, Nature, 479:245-248**). A recent single-molecule study revealed that Rad55-Rad57 binds transiently to Rad51-ssDNA to promote Rad51 filament assembly but then dissociates as the filaments mature (**Roy et al, Mol Cell, 2021, 81:1043-1057**). Therefore, we think the interaction between Bre1 and Rad51-ssDNA could be transient and dynamic. We changed the word “co-complex” to “association or interaction” throughout the Main text (Main text, Page 6, paragraph 3, lines 1, 8 and 9; Page 12, line 3).

References:

1. Liu *et al.*, Rad51 paralogues Rad55-Rad57 balance the antirecombinase Srs2 in Rad51 filament formation. *Nature* **479**, 245-248 (2011).
2. U. Roy *et al.*, The Rad51 paralog complex Rad55-Rad57 acts as a molecular chaperone during homologous recombination. *Mol Cell* **81**, 1043-1057 e1048 (2021).

- Figure 3h: the authors observed the replacement of RAD51 on RPA-coated ssDNA by Bre1. While the increase of RAD51 is apparent, there does not appear to be a corresponding decrease of RPA. Why is that?

Answer: We carefully repeated the experiment with a significantly lower amount of RPA and Rad51. We found that the addition of RPA markedly inhibited Rad51 binding to ssDNA (**Fig.3h, lanes 4 and 7**), while the inclusion of Bre1 in the reaction facilitated Rad51 loading, accompanied by the decrease of RPA binding on ssDNA (**Fig.3h, lanes 7-8**).

- Figure 3h: Preincubation of RPA before RAD51 is added is expected to decrease RAD51 bound to ssDNA. Yet, no decrease is apparent: comparing lanes 4 and 7. Why is that?

Answer: as described above, we repeated the experiment with a lower concentration of RPA and Rad51. Now, we can observe that the addition of RPA markedly inhibited Rad51 binding to ssDNA

(Fig.3h, lanes 4 and 7).

- Assay in Fig. 3l: combination of 7.7 kbp ssDNA and 2.2 kbp dsDNA cannot give rise to nicked DNA after strand exchange.

Answer: We consulted with a leading biochemist expertise in DNA recombination for this technical issue. An alternative and easier assay for measuring strand exchange was recommended (Lee et al, Nature Communications, 10: 65; Lei et al, Nat Communications, 2021,12:6412). In this assay, the Rad51-ssDNA (80-mer oligo) presynaptic filament engages a Cy3-labeled homologous double-strand DNA molecule (40-mer), which is followed by the exchange of strands to yield a labeled DNA joint molecule. Therefore, we performed all the strand exchange experiments with this new assay in the revised manuscript (Fig.3m, 4d, 6i, and 6k).

Reference

Lee CY, *et al.* Promotion of homology-directed DNA repair by polyamines. *Nat Commun*, 2019, 10(1): 65.

Lei KH, *et al.* Crosstalk between CST and RPA regulates RAD51 activity during replication stress. *Nat Commun*. 2021, 12(1):6412.

- Furthermore, in panel 3m, stimulation of strand exchange is expected to enhance the utilization of dsDNA, i.e. the intensity of the dsDNA band should go down when Bre1 is present, which is not at all the case. While I do see an increase in intensity of the strand exchange products, it does not make sense without the corresponding decrease of the DNA substrates. Additionally, ssDNA is already fully engaged even without Bre1.

Answer: As described above, we employed an alternative assay to test the effect of Bre1 on strand exchange. Now, we can see that the addition of Bre1 increases the products of strand exchange, which is accompanied by the reduction in duplex DNA (Fig.3m, lanes 2-6).

- Figure 3d vs SI 6a: In panel 3a, 5 uM of RAD51 did not cause a shift of 100 nM of ssDNA to wells (in fact, only minimal DNA binding was observed). Yet, in panel SI 6a, already 2.5 uM of RAD51 shifted 100 nM dsDNA almost completely to wells. Why is that? The affinities are thought to be similar. Were the substrates of the same length?

Answer: the ssDNA used in Fig.3d is a 7.7-kb circular ssDNA plasmid, while the one used in Extended Data Fig. 6a is a ~1.8-kb linear dsDNA. Although Rad51 has a similar affinity to ssDNA and dsDNA, the differences in DNA size, conformation, the variations in Rad51 binding mode to ssDNA and dsDNA, and the time for electrophoresis likely cause the shift difference.

- In panel 4d, the "nc" form (which is probably not nc, see above) runs below ssDNA, while in panel 3m, it runs above. Why is that?

Answer: In the revision, we used an alternative assay to test the strand exchange, and the new results are presented in Fig.3m and Fig.4d.

- Panel 4d: I am wondering what happened to ssDNA in the presence of Srs2. As above, while I see that Bre1 increases the forms labeled as jm and nc, the proportion of the bands in the experiment does not make a lot of sense.

Answer: This issue was resolved by using the new strand exchange assay. In the new Fig.4d, we can see that the addition of Srs2 attenuates the products of strand exchange (lanes 2-4), while the inclusion of Bre1 could suppress the effect of Srs2 (lanes 5-6).

- The authors conclude that Bre1-Rad51 interaction is required for Bre1 to displace Rad51. However, displacement of Srs2 by Bre1 was observed even in the absence of Rad51 (Fig. 4h). How can the displacement occur when Bre1 and Srs2 do not directly interact without DNA (as noted)?

Answer: This is a very good question. First, our in vitro result showed that Bre1 can directly displace Srs2 from ssDNA in a dose-dependent manner no matter when Bre1 was added together with Srs2 to ssDNA (Fig.4h, lanes 3-6) or when Bre1 was added to the preformed Srs2-ssDNA complexes (lanes 7-9). In both situations, ssDNA is present in the reactions (lanes 4-9). However, Rad51 is not required for Srs2 displacement by Bre1 in vitro.

Notably, our in vivo ChIP results showed that Srs2 loading is increased at DSBs in the *bre1-EK2A* mutant, as seen in the *bre1* deletion or the *bre1-DK2A* mutant (Fig.5c). Based on this result, we conclude that the Bre1-Rad51 interaction is required for Bre1 to displace Srs2 in the chromatin context. Thus, the Bre1-mediated Srs2 displacement from ssDNA has different requirements for Rad51 in vitro and in vivo. We reason that the interaction between Rad51 and Bre1 in vivo might be necessary to bring Bre1 in close proximity to Srs2, allowing the direct Bre1-Srs2 interaction and subsequent Srs2 displacement in the chromatin context.

- Figure 5b: The experiment is a GST pulldown, what is pulled is GST-tagged Srs2. I am confused why there is more Srs2 pulled down in the presence of DNA. Do the authors suggest that pulling on DNA-bound Srs2 may bring other, DNA-bound Srs2 molecules bound to other locations on DNA? Along these lines, Bre1 in the same experiment is bound to Srs2, or independently to other DNA sites? Increasing Bre1 concentrations (comparing lanes 5 and 6) brings down more Bre1 than Srs2, arguing that Bre1 might bind DNA independently.

Answer: Thanks for pointing out the problems. We carefully performed additional GST pull-down assays. The GST-Srs2 pulled down is now even no matter with or without ssDNA. We noted that Bre1 can only be captured by GST-Srs2 in the presence of ssDNA (Fig.5b, lanes 6-7 vs 4-5) but not dsDNA (lanes 8-9). The previous result was replaced with the new Fig.5b.

- Please note whether the interaction patch between Srs2 and Bre1 is conserved (alignment of the

interaction motif).

Answer: We aligned the Srs2-interacting patch of Bre1 with Bre1 homologs from different species, and noted that the residue D484 is not conserved, while the residue K486 is well conserved across species. This result is incorporated in the manuscript (Main text, page 15, paragraph 1, lines 10-13; Extended Data Fig. 15b).

- Fig. 6i: similar concerns as noted above - the total content of DNA in the various lanes does not add up.

Answer: We carried out the strand exchange reaction using the new assay. Now we can see that RNF20 stimulates Rad51-mediated strand exchange in a dose-dependent manner (**Fig.6i**, lanes 2-4). The increase in strand exchange products is accompanied by the decrease in the duplex DNA substrate (Fig.6i, lanes 2-4).

Reviewer #2 (Remarks to the Author):

Homologous recombination (HR) is one of the major double-strand break repair pathways. As the key recombinase, Rad51, once loaded onto single-stranded DNA generated from double-strand break processing, catalyzes homology search and strand exchange. In cells, mediator proteins, such as Rad52 and BRCA2, are required for Rad51 loading due to the RPA barrier. A key aspect of HR research centered upon the assembling and disassembling of the Rad51 nucleoprotein complex. In this manuscript by Liu et al, Bre1, previously known as an E3 ligase functioning in DNA repair, was found to directly interact with both Rad51 and Srs2, a 3'-5' helicase that possesses the anti-recombinase activity. With the separation-of-function mutants isolated that specifically disrupt the identified novel interface, the authors proceeded to apply biochemical and single molecule approaches and conclude Bre1 may function as a new mediator protein to facilitate Rad51 nucleoprotein filament assembling and to antagonizing the anti-recombinase activity from Srs2.

Overall, the data on characterizing the protein-protein interface is solid and the novel functions of Bre1 in assisting Rad51 and antagonizing Srs2 are very interesting. The biochemical data on assessing the mediator activity of Bre1 are, however, less convincing, which, in conjunction with the limited experimental details included in the figure legends makes it very difficult to evaluate the manuscript in its current form. There are a few suggestions depicted below that will help further strengthen the manuscript.

Answer: Thanks for the recognition of our work. We carefully performed additional biochemical experiments and described figure legends in more detail to address all the concerns.

Major points:

1. While some experimental details were provided in the method section, the figure legends were oversimplified. For quite a few experiments, it is not clear what type of substrates were used and at what concentration.

Answer: We now describe the figure legends in more detail, especially the legends of Fig.3, 4, 6, and 7. In addition, we labeled DNA and protein concentrations in more detail in figures or figure legends.

2. The key function of mediator proteins is to assist Rad51-catalyzed strand exchange reaction in the presence of RPA, which competes with Rad51 on ssDNA. In the strand exchange assays presented in Fig 3m, n, and Fig 6i, k, as indicated, 100 nM M13 substrate DNA, 250 nM of Rad51, and 100-170 nM RPA were used. Accordingly, there were on average 2-3 Rad51 molecules and 1-2 RPA molecules per each substrate DNA that is over 7000 nt in length. Under such conditions, both Rad51 and RPA were far from saturating the DNA. With such a concentration of RPA, its barrier effect was probably overlooked, which renders the conclusion of Bre1 as a mediator protein an over-interpretation of the data presented.

Answer: As described above, we now carried out strand exchange reactions using the new assay following the method described by Lee et al. (2019, Nat Commun). In this assay, a short ssDNA (80-mer) and a Cy3-labeled duplex DNA (40-mer) were used as substrates. We observed the stimulatory effect of Bre1/RNF20N on the Rad51-mediated strand exchange reaction in the absence or presence of Srs2/FBH1. The results are presented in the new Fig3m, Fig.4d, Fig.6i-k, and Extended Data Fig.7a.

3. ATP binding and hydrolysis by Rad51 regulate its DNA binding. While ATP was included in the EMSA and Strand Exchange assays, it was not present in the streptavidin pull-down assay based on the buffer conditions provided in the method section.

Answer: Thanks for pointing out this mistake. We included ATP in all ssDNA pull-down assays unless otherwise stated. We now correct the description in the Method section (Main text, Page 28, line 7).

4. Based on the model in the manuscript, a prediction would be the bre1-EK2A and bre1-DK2A mutants may impact the recombination rate. The authors do have a physical assay system in the manuscript (Fig 2d, e), it will be nice to compare the HR rate (crossover vs. noncrossover) in the wild-type, bre1-EK2A, and bre1-DK2A, which will largely strengthen the paper.

Answer: We performed Southern blot analysis to measure the repair kinetics and crossover rate. As expected, we noted that the repair was slower in the *bre1-EK2A* or *bre1-DK2A* mutant cells, and the crossover level was also reduced in the two mutants. The result is presented in the Main text, Page 10, paragraph 3, lines 7-9 and Extended Data Fig. 10a-c. This is consistent with the increased loading of the anti-recombinase Srs2 at DSBs (Fig.5c).

5. In Fig S1, bre1-6A mutation caused an increase in RPA loading, which indirectly affected Rad51

loading. Hence, for the *bre1-EK2A* mutant, a key control experiment that was missing is to examine its impact on RPA loading.

Answer: This is a very good suggestion. We tested the effect of the *bre1-EK2A* mutation on RPA loading in vivo by ChIP. Interestingly, RPA loading was also decreased at DSB ends (**Extended Data Fig.1c**). We think this is likely due to the increased loading of Srs2 at DSBs, since Srs2 is known to be able to evict RPA from ssDNA that allows RPA redistribution and the damping of checkpoint point (Tullino et al, 2017, 21: 570-577, Cell reports; Dhingra et al, PNAS, 2021, 118(8):e2020185118). Thus, the regulation of Srs2 loading at DSBs by Bre1 is important as it can affect the loading of both RPA and Rad51. The defect of Rad51 loading in the *bre1-EK2A* mutant was not due to any increases in RPA loading.

References:

Luisina De Tullio, Kyle Kaniecki, Youngho Kwon, J Brooks Crickard, Patrick Sung, Eric C Greene. Yeast Srs2 Helicase Promotes Redistribution of Single-Stranded DNA-Bound RPA and Rad52 in Homologous Recombination Regulation. *Cell Reports*, 2017, 21(3):570-577.

Nalini Dhingra, Sahiti Kuppa, Lei Wei, Nilisha Pokhrel, Silva Baburyan, Xiangzhou Meng, Edwin Antony, Xiaolan Zhao. The Srs2 helicase dampens DNA damage checkpoint by recycling RPA from chromatin. *PNAS*, 2021, 118(8):e2020185118.

More specific points:

1. In Fig 3b, the molecular weight marker is missing, and it is not clear what percentage of gel was used. Assuming it is a western blot, what antibodies were used to probe?

Answer: We now added the molecular weight markers. The samples were resolved on a 12% SDS-PAGE gel. It is a Western blot analysis with anti-His antibody. It should be noted that both Rad51 and Bre1 used in this experiment are tagged with His-tag.

2. Fig 3 legend states that 50 pM of 90 mer ssDNA was used in the streptavidin pull-down assays, but Fig 3b marked ssDNA as 100 nM.

Answer: Thanks for pointing out this mistake. ssDNA used for streptavidin pull-down assays is 100 nM for Fig.3a and 200 nM for Fig.3h. We corrected the Fig. 3 legend.

3. In Fig 3h, the loading of RPA didn't seem to affect Rad51 binding, why?

Answer: we carefully re-performed the experiment and presented the new results in Fig 3h. Now we can see the presence of RPA markedly reduced Rad51 binding to ssDNA (**Fig.3h, lanes 4 and 7**).

4. In Fig 3m, a control reaction without ATP should be included. Also, the M13 single-strand DNA completely disappeared while only 40% reacted in lane 4, why?

Answer: We performed strand exchange using the new assay and included the control without ATP (Fig.3m, lane 2).

5. In Fig 4g, the result will be clearer if a short ssDNA substrate was used to monitor Srs2 binding without/with Bre1 in the EMSA assay without cross-linking.

Answer: We performed an additional EMSA assay using the M13 mp18 ssDNA without cross-linking, and the result is presented in Fig.4g.

Reviewer #3 (Remarks to the Author):

This article reported a novel function of Bre/RNF20 in HR repair process via regulating the loading of Rad51 to ssDNA. The authors utilized genetic modified yeast strains to prove their hypothesis, and validated some of their results in human cell lines. It is potentially interesting for the field as the findings extend our understanding of how Bre/RNF20 and Rad51 work together in HR repair regulation. Although they discussed possible mechanisms, however, this manuscript lacks explanation of how Bre1/RNF20 regulates the loading and unloading of proteins such as Rad51, Srs2 and RPA, since Bre/RNF20 itself has no binding to ssDNA.

Answer: Thanks to the reviewer for finding our work of interesting and significance. In this study, we focus on the multifaceted functions of Bre1/RNF20 as a recombination mediator, including their roles in stimulating Rad51-ssDNA assembly, replacing RPA, inhibiting dsDNA binding, promoting RAD51 strand exchange, and in antagonizing Srs2/FBH1 helicases. As reviewer #1 mentioned, we have already included a large amount of work in this manuscript. How Bre1/RNF20 regulates the loading and unloading of Rad51, Srs2 or RPA are certainly interesting questions. Currently, we are collaborating with structural biologists, aiming to address the mechanisms using the structural biology approach. We hope to present the results soon in separate studies.

Moreover, most of the conclusions are based on either in vitro assay or overexpression, especially the results in human cells. For instance, the interaction assay for Rad51/RNF20 and FBH1/RNF20. In addition, experiment such as immunostaining or pull-down assay with genetic modified (KO or KD) cell lysates should be performed to show the loading or unloading of related proteins to DSB in vivo. Furthermore, whether the effects of Bre/RNF20 on HR repair are mediated by Rad51 should be clarified (using Rad51 KO or KD cells to detect the effects of RNF20 on HR).

Answer: Thanks for the suggestions. We apologize for that the previous Fig.6a was mislabeled. Actually, it was an immunoprecipitation assay carried out with endogenous proteins. Now, we corrected the labeling in Fig.6a. In addition, we performed immunoprecipitation with an anti-RNF20 antibody to test the endogenous RNF20-FBH1 interaction, and the result is presented in the Main text, Page 12, paragraph 3, lines 6-8, and Fig.6m (bottom panel).

The effect of RNF20 on the loading and unloading of other HR proteins, such as, BRCA1, RPA, and RAD51, has been well documented in several previous studies (Nakamura et al, Mol Cell,

41:515-528, 2011, Fig.1 and Fig.S1; Moyal et al; Mol Cell, 41:529-542, 2011, Fig.6 and Fig.S5). Therefore, here, we just added BRCA1 foci formation in addition to RAD51 foci formation to support our in vitro result. The result is presented in the Main text, page 13, paragraph 3, lines 1-3 and page 14, line 1; Extended Data Fig.13c,d.

To test whether the effect of RNF20 on HR repair is mediated by Rad51, we measured HR efficiency using the DR-GFP reporter. Knockdown RNF20 impaired HR repair, while the depletion of RAD51 attenuated HR repair to a greater extent. Additional depletion of RNF20 in RAD51 knockdown cells did not further reduce HR efficiency. This result suggests that the role of RNF20 in HR repair is completely dependent on RAD51. This result was incorporated into the manuscript (Main text, page 14, lines 4-6 and Extended Data Fig.14c,d).

Minor points:

Why the two amino acids are only important for some proteins (RPA, but not Srs2) unloading. What could be the possible explanation?

Answer: We would assume that the reviewer is asking why bre1-EK is important for the unloading of RPA but not Srs2. Actually, the two amino acids EK are important for the unloading of both RPA and Srs2 in vivo, while they are only required for the unloading of RPA but not Srs2 in vitro. The two amino acids mediate the Bre1-Rad51 interaction. They are important for Bre1 to stimulate Rad51 assembly on ssDNA, which could directly drive RPA unloading in vitro and in vivo.

Our ChIP results revealed that the amino acids EK are also required for Bre1 to unload Srs2 in vivo, since the bre1-EK2A mutation leads to increased Srs2 loading at DSBs (Fig.5c). In contrast, the EK are not required for Bre1 to displace Srs2 from ssDNA in vitro (Extended Data Fig.9c-d, lanes 3-5). We reason that the Bre1-Rad51 interaction might be necessary to bring Bre1 in close proximity to Srs2, allowing the direct Bre1-Srs2 interaction and subsequent Srs2 displacement in the chromatin context. However, the assess of Bre1 to Srs2 in vitro only requires ssDNA but not Rad51. Therefore, the bre1-EK2A mutant protein could still displace Srs2 in vitro.

The authors claimed that Bre1 promotes the replacement of ssDNA-bound RPA by Rad51, but why the amount of PRA pull-down by ssDNA was not changed in line 7 and lane 8 of Fig.3h? The same issue was observed in Fig.6f, where no decrease of ssDNA-bound RPA with the addition of RNF20N.

Answer: We carefully performed additional ssDNA pull-down experiments and the new results are present in Fig.3h and Fig.6f. Now, we can observe the stimulatory effect of Bre1 or RNF20 on Rad51 replacement of RPA. The increased binding of Rad51 on ssDNA is accompanied by reduced binding of RPA (Fig.3h, lanes 7-8; Fig.6f, lanes 5-8).

The total amounts of ssDNA and dsDNA in Fig3m should be the same, but why the lane 5,6 have more dsDNA left, although they have more jm and nc in the meantime.

Answer: We performed strand exchange reactions using an alternative assay and the new data is presented in **Fig.3m**.

How the authors did the normalization in **Fig.3o**? the amount of jm for sure is not increased with condition of pep104 in 500 nM.

Answer: This result was removed from the revised manuscript.

What is the difference of lane 6,7 and lane 4, 5 of Fig.S6a? How the authors make the conclusion that Bre1 could not remove Rad51 from the pre-formed Rad51-dsDNA complex?

Answer: For the samples in **Extended Data Fig.6a, lanes 4-5**, Bre1 and Rad51 were added at the same time to dsDNA, while for the samples in **lanes 7-8**, Rad51 was preincubated with dsDNA for 10 mins to allow the formation of Rad51-dsDNA prior to the addition of Bre1. Our results suggest that Bre1 restrains Rad51 binding to dsDNA, yet it cannot displace the Rad51 that has already bound on dsDNA.

Will dsDNA mediate the interaction of Srs2 and Bre1 (Fig.5b)?

Answer: We repeated the experiment and included dsDNA. The result indicates that only ssDNA, but not dsDNA, mediates the interaction between Srs2 and Bre1 (**Fig.5b, lanes 6-7 vs 8-9**).

According to the text, it should be siRNA instead of shRNA in Fig.7a,7c.

Answer: Thanks for pointing out this mistake. It is shRNA. We fixed it in the text (**Main text, Page 13, paragraph 2, lines 9-13**).

Other issues: Manuscript needs to be revised carefully to avoid errors: Streptavidin instead of Streptavadin in Fig.4a; “p” or “p”.

Answer: It has been corrected throughout the text and figures.

REVIEWER COMMENTS

Reviewer #1 (Remarks to the Author):

I thank the authors for their efforts on revising the manuscript, most of the points have been addressed in revision. I have a few questions regarding the new strand exchange assays:

- Magnesium concentration used was very low (0.2 mM), which could lead to artifactual products mediated by DNA melting and annealing and not strand exchange activity. This melting would be dependent on ATP, as ATP would remove last of the the free magnesium. Lane 7 in 4d is a good control that indicates that this is likely not the case. Nevertheless, why was such low [Mg] chosen for the assays? Does Bre1 stimulate even without RPA? The contrast of the strand exchange assays images is too high, please reduce it.

Reviewer #2 (Remarks to the Author):

My previous concerns have been properly addressed. The newly performed experiments have further strengthened this very interesting work.

Reviewer #3 (Remarks to the Author):

I only have a major comment. How do the authors explain the inconsistent results of the same experiment in different times? For example, the Fig.3h in different versions of the manuscript. In their first version of manuscript, Bre1 has no effect on the replacement of RPA. Surprisingly, Bre1 strongly promotes this replacement. Why?

A point-by-point response to the reviewer's questions

Reviewer #1 (Remarks to the Author):

I thank the authors for their efforts on revising the manuscript, most of the points have been addressed in revision. I have a few questions regarding the new strand exchange assays:

- Magnesium concentration used was very low (0.2 mM), which could lead to artifactual products mediated by DNA melting and annealing and not strand exchange activity. This melting would be dependent on ATP, as ATP would remove last of the free magnesium. Lane 7 in 4d is a good control that indicates that this is likely not the case. Nevertheless, why was such low [Mg] chosen for the assays? Does Bre1 stimulate even without RPA? The contrast of the strand exchange assays images is too high, please reduce it.

Answer: We apologize that the Magnesium concentration (0.2 mM) described in the previous Methods was a typing error. Actually, we used 2 mM Mg²⁺ in the strand exchange reactions, as described by Krejci et al. (Nature, 2003, 423, 305–309). The protocol described by Lei et al. used 1 mM Mg²⁺ in the strand exchange reactions (Nat Communications, 2021,12:6412). We corrected the error in the revised manuscript (Main text, Page 29, paragraph 3, line 2). In addition, we performed all the strand exchange reactions in the presence of RPA. It is unlikely Bre1 can stimulate the Rad51-mediated strand exchange without RPA. As suggested, we have adjusted the contrast of the strand exchange assay images in the revised manuscript (**Figure 3m, 4d, 6i, 6k**).

References

Krejci, L., Van Komen, S., Li, Y., Villemain, J., Reddy, M.S., Klein, H., Ellenberger, T., and Sung, P. (2003). DNA helicase Srs2 disrupts the Rad51 presynaptic filament. *Nature* 423, 305-309.

Lei KH, et al. Crosstalk between CST and RPA regulates RAD51 activity during replication stress. *Nat Commun.* 2021, 12(1):6412.

Reviewer #2 (Remarks to the Author):

My previous concerns have been properly addressed. The newly performed experiments have further strengthened this very interesting work.

Reviewer #3 (Remarks to the Author):

I only have a major comment. How do the authors explain the inconsistent results of the same experiment in different times? For example, the Fig.3h in different versions of the manuscript. In their first version of manuscript, Bre1 has no effect on the replacement of RPA. Surprisingly, Bre1 strongly promotes this replacement. Why?

Answer: Thanks for the comment. We tested the effect of Bre1 on the replacement of ssDNA-bound

RPA by Rad51 using ssDNA pull-down assays. In the former experiment, 200 nM biotinylated oligonucleotide (90 nt) was premixed with a higher concentration (1.8 μM) of RPA, which is followed by the addition of 5 μM of Rad51 and 1 μM of Bre1. We observed that the addition of Bre1 stimulates Rad51 loading on ssDNA, while we did not see a corresponding decrease of RPA binding (previous Fig.3h, lane 7-8). We think that the experiment was likely carried out in a suboptimal condition. To resolve this issue, we repurified RPA, Rad51, and Bre1 during the revision and optimized the stoichiometric ratio of Bre1, Rad51, and RPA in the reaction. We then carefully repeated the experiment with a significantly lower amount of RPA (200 nM) and Rad51 (0.6 μM). Under this condition, we found that the inclusion of Bre1 (1.2 μM) in the reaction facilitated Rad51 loading, accompanied by the decrease of RPA binding on ssDNA (New Fig.3h, lanes 7-8). The role of Bre1 in promoting RPA replacement by Rad51 was also confirmed by the single-molecule studies (Fig.3k). We think the concentration, activities, and stoichiometric ratio of these proteins are the key to the success of these biochemical experiments. That might explain the differences.